



# A zero-dimensional view of atmospheric degradation of levoglucosan (LEVCHEM_v1) using numerical chamber simulations

Loredana G. Suciu[1], Robert J. Griffin[2], Caroline A. Masiello[1,3]

[1]Department of Earth, Environmental and Planetary Sciences, Rice University, Houston, 77005, USA
[2]Departments of Civil and Environmental Engineering and Chemical and Biomolecular Engineering, Rice University, Houston, 77005, USA
[3]Departments of Chemistry and Biosciences, Rice University, Houston, 77005, USA

*Correspondence to*: Loredana G. Suciu (lgs4@rice.edu)

**Abstract.** Here we developed a zero-dimensional (0-D) modeling framework (LEVCHEM_v1) to provide insights into the
atmospheric degradation of a key tracer emitted during biomass burning - levoglucosan (LEV), while additionally exploring its effects on the dynamics of secondary organic aerosols (SOA) and other gases. For this, we updated existing chemical mechanisms (homogeneous gas-phase chemistry and heterogeneous chemistry) in the BOXMOXv1.7 model to include the chemical degradation of LEV and its intermediary degradation products in both phases (gas and aerosol). In addition, we added a gas-particle partitioning mechanism to the model to account for the effect of evaporation and condensation on the
phase-specific concentrations of LEV and its degradation products. Comparison of simulation results with measurements from various chamber experiments show that the degradation time scale of LEV varied by phase, with degradation occurring over ~1.5-3.5 days and aerosol-phase degradation occurring over ~8-21 hours. These relatively short time scales suggest that most of the initial LEV concentration can be lost chemically or deposited locally before being transported regionally. We varied the heterogeneous reaction rate constant in a sensitivity analysis and found that longer degradation time scales of LEV
are possible in the gas phase (5 days) and the aerosol phase (7 days), implying that some LEV may be transported regionally. The multiphase chemical degradation of LEV has effects on SOA and other gases. Several first- or second-generation products resulted from its degradation; most of the products include one or two carbonyl groups, one product contains a nitrate group, and a few products show the cleavage of C-C bonds. The relative importance of the products varies depending on the phase and the timing of the maximum concentration achieved during the simulation. Our estimated secondary organic
aerosol SOA yields (5-32%) reveal that conversion of LEV to secondary products is significant and occurs rapidly in the studied scenarios. LEV degradation affected other gases by increasing the concentrations of radicals and decreasing those of reactive nitrogen species. Decreases of the mixing ratios of nitrogen oxides appear to drive a more rapid increase in ozone compared to changes in volatile organic compounds (VOC) levels.

An important next step to confirm longer degradation time scales will be to extend the evaluation of the modeled LEV
degradation beyond 3-5 hours, by using more extensive data from chambers, and, possibly from fire plumes. The mechanism developed here can be used in chemical transport models applied to fire plumes to trace LEV and its degradation products





from source to deposition, assess their atmospheric implications and answer questions relevant to fire tracing, carbon and nitrogen cycling, and climate.

## 1 Introduction

Knowledge of the atmospheric lifetimes of biomass burning emissions is critical to predict their impacts on photochemistry, air quality and climate. The organic compounds in these emissions are denoted as pyrogenic carbon (PyC) and together they cover a wide range of chemistries and phases, making the determination of individual lifetimes challenging. In the atmosphere, PyC can be in the condensed phase (predominantly as semi- and non-volatile particulate matter, PM) and/or in the gas phase (volatiles). Both phases participate in atmospheric photochemistry. For instance, volatile organic compounds

(VOC) react with hydroxyl radical (OH) and contribute to tropospheric ozone ($O_3$) formation. Other gases released during biomass burning, such as polycyclic aromatic hydrocarbons, can be oxidized, the products of which may form semi- or non-volatile PM. Both directly emitted and secondarily formed PM alters visibility (through light extinction), human health (through respiration) and climate forcing (via absorption/scattering of solar radiation). Depending on its chemical and physical properties, PM also participates in cloud formation as cloud condensation nuclei and influences the physics and

chemistry of clouds. Through alterations of physical properties of clouds, PM indirectly contributes to climate forcing. The magnitude and the extent of PyC impacts depend on its atmospheric lifetime.

Anhydrosugars, the most abundant of which is levoglucosan (LEV), are molecular tracers of PyC that traditionally have been used as markers for biomass burning in ambient aerosols, or as markers for wildfires in sediments and ice cores (Suciu et al., 2019 and references therein). However, their degradation and lifetimes are not well understood in any environment,

including the atmosphere and cryosphere, two environments that are related via atmospheric transport and deposition of such PyC tracers. Therefore, understanding the atmospheric fate of anhydrosugars is essential not only to understanding fire effects on air quality but also to interpreting fire records in ice, and to studying the complex relationship between fire, vegetation and climate.

Experimental laboratory studies (in chambers or flow tubes) on LEV chemical degradation suggest that its atmospheric

lifetimes vary widely, from minutes to months (Hennigan et al., 2010; Hoffman et al., 2010; Kessler et al., 2010; Knopf et al., 2011; Bai et al., 2013; Lai et al., 2014; Slade and Knopf, 2014; Arangio et al., 2015; Gensch et al., 2018). In addition, the multiphase character of LEV and its gas-particle partitioning (G/P) between phases has not been explicitly considered yet in laboratory studies of its chemical kinetics. Given its semi-volatile nature, the evaporation/condensation effect in conjunction with chemical kinetics must be given attention in the estimation of LEV lifetimes, especially those with respect to chemical

degradation. Some models, such as the non-equilibrium kinetic evaporation model of May et al. (2013) consider this.

To estimate more accurately the atmospheric degradation time scales (modelled decay of concentration over time relative to initial concentration), anhydrosugar chemistry must be studied in more complex atmospheric settings than those reproduced in the laboratory. This could be achieved using three-dimensional (3-D) chemical transport models (CTMs). However,





current CTMs do not treat anhydrosugars individually in their chemical mechanisms. This is partly because these models

often are motivated by the need to quantify only PM mass to meet air quality legislation. Thus, studies often report modeled species such as $PM_{2.5}$ (that with diameters smaller than 2.5 microns), organic carbon in PM (OC), and black carbon (BC) (In et al., 2007; Alvarado et al., 2009; Simon and Bhave, 2012; Pye and Pouliot, 2012; Heron-Thorpe et al., 2014; Alvarado et al., 2015). Moreover, because anhydrosugars are also semi-volatile they participate in both gas and aerosol-phase chemistries, so placing them into just one single category (i.e., $PM_{2.5}$) is inaccurate. In general, individual emissions from

biomass burning are lumped into categories, assuming that all species behave identically with respect to chemical and physical transformation or loss. While this assumption eases the computational burden of the chemistry and physics of the model, it can yield inaccurate results regarding the modeled species; it also does not allow the study of tracers individually.

Here we developed use a zero-dimensional (0-D) modeling framework (LEVCHEM_v1) to study the chemical degradation of LEV. Because the two isomers of LEV (mannosan and galactosan) have similar structures but different arrangements of

the hydroxyl groups, this study only focuses on chemical reactions involving LEV. A future goal is to expand LEVCHEM_v1 to include the degradation of the two isomers and, then, to implement the full mechanism of anhydrosugar degradation into 3-D CTMs. The 0-D modeling approach here can identify model uncertainty attributable to the mechanism only; when the mechanism is used in a CTM, other sources of uncertainties (advection, diffusion, deposition, etc.) in the overall uncertainty of the model predictions can be assessed.

Several research topics pertinent to the chemical degradation of LEV are dealt with in this study. These will be addressed after a discussion of the model framework and development.

First, we explore the degradation time scale of LEV, and what can be inferred form it regarding the scale of its impact (local versus regional). For example, isolating the effect of chemistry from transport or other physical processes may yield different degradation time scales, resulting in different inferred transport distances, impacting whether local- or regional-scale

chemistry may be the dominant process controlling the lifetime of LEV.

Second, we examine the contribution of LEV degradation to the formation of secondary organic aerosols (SOA), including changes in total PM mass and the relative importance of degradation products. Significant LEV degradation may lead to higher SOA yields. This information can further be used as a reference to understand SOA formation in a 3-D CTM framework.

Third, we examine how LEV degradation affects the concentrations of other gases such as $O_3$ and its precursors, nitrogen oxides ($NO_x$ = nitric oxide (NO) + nitrogen dioxide ($NO_2$)) and VOC, total reactive nitrogen ($NO_y$) and $NO_x$ oxidation products ($NO_z = NO_y - NO_x$). Considering its multiphase chemistry that also generates peroxy radicals ($RO_2$), LEV may have an important effect on these pollutants.





## 2 Modeling approach

### 2.1 Overview of the 0-D modeling framework and mechanisms

The 0-D model used to develop LEVCHEM_v1 in this study (BOXMOX v1.7) (Knote et al., 2015) is a publicly available software that expands on earlier code, the Kinetic Pre-Processor (KPP v2.1) (Sandu and Sander, 2006). The two models are briefly described below.

The KPP generates code using chemical reactions and their respective reaction rate coefficients as inputs (Sandu and Sander, 2006). The rate of change in concentration of a species $i$ ($\frac{dCi}{dt}$) is expressed as the difference between its production (P) and loss (L) rates (eq. 1).

$$\frac{dCi}{dt} = P - L \tag{1}$$

The generated code (which determines the $P$ and $L$ terms in eq. 1) then is used in a temporal integration to compute the change in concentration of the individual reactants and products based on a system of ordinary differential equations (ODE). The KPP offers a variety of stiff numerical integrators that can be selected by the user in order to maximize the computational efficiency of the ODE system within a low to medium accuracy regime (Sandu and Sander, 2006).

The BOXMOX extends the KPP capabilities even further by providing a framework in which various numerical experiments are possible, such as chamber experiments or boundary layer atmospheric chemistry numerical experiments (Knote et al., 2015). These are possible with the addition of a wrapper to the KPP. The wrapper allows the user to add inputs to the model, such as initial conditions, environmental conditions, boundary conditions, time-varying photolysis rates, turbulent mixing, emissions, deposition, etc., in order to run numerical experiments; it also allows the user to add new code to further develop the model. The model outputs time series of species concentrations, rate constants and other user-specified outputs.

### 2.2 Mechanistic development

We integrated the multiphase chemical degradation of LEV into BOXMOXv1.7 by adding chemical reactions along with their reaction rate coefficients to existing homogeneous gas-phase and heterogenous mechanisms (LEVCHEM_v1). These existing mechanisms already have been implemented and tested by the BOXMOXv1.7 developers (Knote et al., 2015).

Based on its similarity to mechanisms used in 3-D CTMs, we chose the Carbon Bond version 2005 with Toluene Updated Chlorine Chemistry as the homogeneous gas-phase mechanism to implement the gas-phase degradation of LEV. This was recently updated by the U.S. EPA to include additional tropospheric chemistry (CB05TUCl_EPA). It contains 148 chemical reactions that constitute the core of the mechanism or the "CB05" part (Yarwood et al., 2005; Whitten et al., 2010), 23 reactions for the reactive chlorine chemistry or the "TUCl" part, 10 reactions for formation of secondary aerosols from gas-gas reactions, and 24 photolysis reactions (Knote et al., 2015). In total, the overall gas-phase mechanism included 205 reactions and 82 variable species to describe gas-phase tropospheric photochemistry. Here we extended the CB05TUCl_EPA mechanism to include 13 reactions and 10 species (radicals and 1st or 2nd generation products) associated with LEV





chemistry in the gas phase (see Table 1). (Chemical structures are shown in Figure S1 and Figure S2 in the Supplemental

Information). Thus, the total number of reactions and species in the updated gas-phase mechanism increased to 218 and 99,

respectively.

The homogeneous gas-phase reaction rate coefficients (Table 1) were modeled as constants (when available in the literature)

or as Arrhenius type reaction rate coefficients (eq. 2) using functions developed previously (Knote et al., 2015) with

measured, assumed or calculated parameters:

$$k = A * exp\left(\frac{-E}{RT}\right) \tag{2}$$

where $k$ is the homogeneous second-order gas-phase reaction rate coefficient (cm$^3$ molecules$^{-1}$ s$^{-1}$), $A$ is the collision

frequency factor (cm$^3$ molecules$^{-1}$ s$^{-1}$), $E$ is the energy barrier for the reaction (kJ mol$^{-1}$), $R$ is the ideal gas law constant

(8.314 J mol$^{-1}$ K$^{-1}$) and $T$ is temperature (K).

When the collision rate coefficient $A$ was not available in the literature, we calculated it using eq. 3 (Seinfeld and Pandis,

2006) applied to two spherical bodies (molecules) A and B:

$$A = \pi d^2 \sqrt{\left(\frac{8k_BT}{\pi\mu}\right)} \tag{3}$$

where $d^2$ represents the squared sum of the two radii of A and B (m$^2$) while the term under the square root is the relative

velocity of the A and B collision bodies in which $k_B$ is the Boltzmann constant (1.381 x 10$^{-23}$ J K$^{-1}$) and μ is the reduced mass

(eq. 4):

$$\mu = \frac{m_A * m_B}{(m_A + m_B)} \tag{4}$$

The heterogeneous chemical mechanism HETCHEM was developed by Knote et al. (2015) to model the heterogeneous

interaction between dinitrogen pentoxide (N$_2$O$_5$) and water bound to solid aerosols or PM. The heterogeneous reaction rate

($k_{SFC\_REACTION}$) was modeled by Knote et al. (2015) based on first-order surface uptake from Fuchs and Sutugin (1971) (eq.

145    5):

$$k_{SFC\_REACTION} = \frac{1}{4} * \gamma * \omega * SAD \tag{5}$$

where γ represents the uptake coefficient of the gas-phase oxidant species $i$ (ranging from 0 to 1), ω is the mean molecular

velocity (m s$^{-1}$) and SAD is the aerosol surface area or surface area density (m$^2$ m$^{-3}$). The mean molecular velocity is

calculated via eq. 6:

$$\omega = 1.455 * 10^2 * \sqrt{\frac{T}{MW * 10^3}} \tag{6}$$

where $T$ is the temperature (K) and MW is the molecular weight of the gas species (kg mol$^{-1}$).

Using the same expression for the heterogeneous reaction rate as in eq. 5, we implemented the heterogeneous chemistry of

LEV in the form of 1$^{st}$ order reactions (see Table 2) and using uptake coefficients (γ) available from literature (experimental

measurements) or calculated in this study based on the collision theory (Seinfeld and Pandis, 2006), thermodynamic

parameters from Bai et al. (2013), and the relationship between $\gamma$ and the second-order heterogeneous reaction rate constant





for the reaction of LEV with the OH radical (Kessler et al., 2010). When the uptake coefficient was not available in the literature, eq. 7 was used to calculate the uptake coefficient for the heterogeneous reaction of particle-phase LEV (and its degradation products):

$$\gamma_{i,OH} = \frac{2D_0 \rho_i N_A}{3\bar{c}_{OH} M_i} k_{i,OH} \tag{7}$$

where $\gamma_{i,OH}$ is the effective gas-phase oxidant uptake coefficient by species $i$ (here, the gas-phase oxidant being OH), $D_0$ is the surface-weighted average diameter of the particle at the beginning of the experiment (in this study, the particle diameter was assumed to be constant throughout the simulations and is denoted as $D_p$), $\rho_i$ is the density of the organic species, $N_A$ is Avogadro's number (6.022 x $10^{23}$ molecules mol$^{-1}$), $\bar{c}_{OH}$ represents the average velocity of the gas-phase OH radical (or other oxidant), $M_i$ is the molecular weight of the organic species and $k_{i,OH}$ is the second-order heterogeneous reaction rate

constant. This study used an average of several heterogeneous reaction rate constants (2.85 x 10$^{-13}$ cm$^3$ molec$^{-1}$ s$^{-1}$) measured by Slade and Knopf (2014). We assumed this value for all LEV degradation products (including for the radical LEVROOH, see reaction 9 in Table 2) due to the fact that experimental heterogeneous reaction rate coefficients have not been measured for LEV products.

The G/P mechanism used in this study (as part of LEVCHEM_v1) was taken from May et al. (2013) and describes the rate

of change in concentration of both gas-phase and particle-phase species due to evaporation and condensation (eq. 8 and 9).

$$\frac{dC_{p,i}}{dt} = -CS(X_{m,i} Ke_i C_i^* - C_{g,i}) \tag{8}$$

$$\frac{dC_{g,i}}{dt} = -\frac{dC_{p,i}}{dt} \tag{9}$$

Changes in the particle phase concentration ($C_{p,i}$) are tracked simultaneously based on the difference between the gas-phase concentration of species $i$ ($C_{g,i}$) and the theoretical surface equilibrium concentration ($C_i^*$) (eq. 10), corrected for the mass

fraction of species $i$ in the particle phase ($X_{m,i}$) (eq. 11) and the Kelvin effect ($Ke_i$) (eq. 12):

$$C_i^*(T) = C_i^*(298\ K) exp\left[-\frac{\Delta H_{vap,i}}{R}\left(\frac{1}{T} - \frac{1}{298\ K}\right)\right]\frac{298\ K}{T} \tag{10}$$

where $C_i^*(298\ K)$ represents the saturation concentration of species $i$ at 298 K and $\Delta H_{vap,i}$ is the enthalpy of vaporization of species $i$. From a mass balance, the changes in the two concentrations are equal but opposite in sign.

$$X_{m,i} = \frac{f_i C_{tot}}{C_{OA}}\left(1 + \frac{C_i^*(T)}{C_{OA}}\right)^{-1} \tag{11}$$

where $f_i$ represents the mass fraction of the organic species $i$, $C_{tot}$ is the total concentration of the organics (gas and aerosol phases) and $C_{OA}$ is the total concentration of organic aerosols.

$$Ke = \frac{4\sigma MW_i}{\rho RT D_p} \tag{12}$$

where $\sigma$ represent the surface tension of the bulk particle, $MW_i$ is the molecular weight of the organic species $i$, $\rho$ is the bulk density of the particle and $D_p$ is the particle diameter.





The first order condensation sink (CS) (eq. 13) is a function of $D_p$, total particle number concentration ($N_t$), the diffusion coefficient of the organic vapor in air ($D$) and the Fuchs-Sutugin correction factor ($C_{F-S}$) that accounts for effects of non-continuity (eq. 14).

$$CS = 2\pi D_p N_t D C_{F-S} \qquad (13)$$

The $C_{F-S}$ depends on the Knudsen number ($Kn$) and the mass accommodation coefficient ($\alpha$).

$$C_{F-S} = \frac{1+Kn}{1+0.3773Kn+1.33Kn\frac{1+Kn}{\alpha}} \qquad (14)$$

The dimensionless Knudsen number (eq. 15) is defined as the ratio between the mean free path of organic molecules in air ($\lambda$ = 62.5 nm) and the particle radius ($D_p/2$).

$$Kn = 2\frac{\lambda}{D_p} \qquad (15)$$

The mass accommodation coefficient represents the probability of a vapor sticking to the particle surface once a collision

occurs; numerically, $\alpha$ ranges from 0 (no accommodation) to 1 (perfect accommodation) (Seinfeld and Pandis, 2006).

**2.3 Simulations and sensitivity analysis**

For both model evaluation and sensitivity, we ran multiple 7-day simulations at 10-second temporal resolution in various scenarios, from fast (the default case) to relatively slower heterogeneous chemistry. The heterogeneous chemistry was varied to account for other controls on LEV concentration that were not explicitly considered in the present 0-D modeling

approach, such as aerosol matrix effects (composition, mixing state, multilayer kinetics, liquid water content, etc.). These additional controls were lumped into a single factor (F) which, for model evaluation, was assumed to vary according to the conditions in chamber experiments. For sensitivity analysis, we varied F from 1.0 (default case) to lower values (0.1, 0.01 and 0.001) to slow down the heterogeneous reaction rates. In addition, we varied the mass accommodation coefficient (see eq. 14) from a default case (which is 0.1 for a system in equilibrium (May et al., 2013)) to lower values (0.01 and 0.001).

The mass accommodation coefficient is related to the G/P partitioning mechanism.

The initial conditions of aerosol-phase LEV represent the average of initial concentrations used in chamber experiments (Hennigan et al., 2010; Lai et al., 2014) (Table S1 in the Supplemental Information). The initial LEV concentration in the gas phase was set to its vapor pressure in all the scenarios (Table S1). We estimated the initial conditions of other species in the chemical mechanism as well as photolysis rate constants by running 1-h resolution of daily 3-D CTM simulations

(Community Multi-scale Air Quality Model, CMAQv5.0.2) using inputs (emissions and meteorology) from Rasool et al. (2016). These conditions correspond to the location, altitude, and timing of a small prescribed-fire plume in South Carolina (Sullivan et al., 2014).

For one aerosol property ($D_p$), air temperature, pressure and relative humidity values, we used values from chamber experiments (Hennigan et al., 2010; Lai et al., 2014) (Table S1). Other parameters ($N_t$, $SAD$, $\Delta H_{vap,i}$, $\sigma$, $C_i^*(298\ K)$ and $\rho$)

that were not measured in chamber experiments but were used in simulations are also given in Table S1.





# 3 Results and discussion

## 3.1 Model evaluation

We evaluated the model (LEVCHEM_v1) by comparing simulation outputs (i.e. concentration) with experimental chamber data in scenarios in which simulations were initialized using chamber conditions. In particular, we investigated the
contributions of LEV degradation to SOA, the change in total PM mass and the effects on other gases like $O_3$ and $NO_x$. We also examined the sensitivity of the degradation time scale of LEV and SOA yields to model parameters.

We evaluated the two-phase (gas-aerosol) modeling of LEV degradation by comparing the time-series of aerosol LEV concentration resulting from simulations to those obtained from laboratory chamber experiments (only the particle phase data) over 5 hours (Figure 1). Overall, the model predicted that LEV degradation closely follows the measured LEV
degradation in relatively slower heterogeneous chemistry scenarios (F = 0.002; 0.004; 0.02; 0.03, depending on the experimental data considered) and at low mass accommodation coefficient (0.001). The one order of magnitude difference between F values may be explained by the different initial LEV concentration used in both experiments and simulations (which is one order of magnitude as well) and, to a smaller extent, by the differences in relative humidity (Table S1). For instance, Hennigan et al. (2010) used drier conditions in chamber experiments compared to Lai et al. (2014). However, the
model does not capture fast degradation in one case (red dots) in the first hour of simulation and the plateau observed after three hours (diamonds and triangles). While the first case may be explained by the uncertainty in the modeled heterogeneous reaction rate that is varied by F, the second case could be explained by the fact that, in chamber experiments, the build-up of matter at the surface of the aerosol prevents LEV in the aerosol reacting with gases or partitioning to the gas phase. The scattering in the chamber data relative to model lines could also be explained by the different source of LEV used in
chamber experiments compared to the model (wood smoke particles and smoke extract *versus* pure LEV particles).

One-to-one comparison of predicted versus measured LEV degradation (Figure 2) from all the simulated scenarios (red and blue) shows that the model performs very well for some of the data points (those that fall within the ± 30% limits) but the average absolute error of the model is relatively large (48%). Overall, the model underpredicts the LEV concentration (average relative error of -48%). The linear agreement between the model predictions and the experimental data is strong
(coefficient of determination of 0.78).

While only the first five hours of the simulations could be evaluated using chamber measurements, the simulated LEV degradation continued after this length of time until LEV concentration was zero, 1.5-3.5 days in the gas phase and 8-21 hours in the aerosol phase (Figure 3). These longer time scales are a first estimate of degradation time scales of LEV.

The relative importance of degradation products differs in the two phases (see Table 1 and Table 2 for processes leading to
formation of these products; also see Figure S1 and Figure S2 in the Supplemental Information for chemical structures), with LEVP4 and LEVP5 dominating the gas phase and LEVP6, LEVP7 and LEVP2 dominating the aerosol phase over the first 5 hours (Figure S3 to Figure S4 in Supplemental Information). LEVP4 is a product formed only by the gas-phase chemistry (reaction 6 in Table 1) and contains a carbonyl group after this reaction (Figure S1 in Supplemental Information). LEVP5 is



a nitrated organic (Figure S1 in Supplemental Information) that is theoretically generated by both chemical mechanisms
(reaction 13 in Table 1 and reaction 14 in Table 2). Products LEVP6 and LEVP7 (Figure S1 in Supplemental Information)
are results of the fragmentation pathway specific only to heterogeneous chemistry (reactions 10-11 in Table 2); they both
contain a carbonyl group (Figure S1 in Supplemental Information). LEVP2 is a product of reaction 4 (Table 1) and reaction
5 (Table 2); it contains two additional functional groups compared to LEV: a carbonyl and an ether (Figure S1 in
Supplemental Information). The relative importance of products slightly changes beyond 5 hours, particularly in the aerosol
phase, in which LEVP3 becomes more important than LEVP2. LEVP3 is the largest molecular product (Figure S1 in
Supplemental Information) that is generated by the multiphase LEV chemistry in reactions 5 (Table 1) and 6 (Table 2).
Through subsequent reactions, LEVP3 can grow into a larger molecule that would ultimately contribute to the nucleation of
new PM (Bai et al., 2013).

Decreasing the rate of heterogeneous chemistry by one order of magnitude  has little effect on the relative importance of
products (Figure S5 to Figure S6 in Supplemental Information), While this shifts the timing of the maximum concentration
in both phases, it has a more important effect on concentration of the product, becoming lower in the aerosol phase and
higher in the gas phase, suggesting that evaporation dominates over heterogeneous chemistry. The concentrations of
products in the aerosol phase are all drivers of SOA yields; it is thus important to know the main molecular SOA
composition resulting from LEV degradation and G/P partitioning.

### 3.2 Contribution of levoglucosan degradation to SOA

Traditionally, reactant organic species in the gas phase are considered to contribute to new SOA formation (or new SOA
mass). However, in this study, since LEV is present in both phases and its chemistry generates products in both phases that
can partition from one phase to another, both LEV_G (gas) and LEV_A (aerosol) can be treated as SOA precursors. Thus,
they are both included in SOA yield calculation. Using eq. 16, the SOA yield is calculated as the ratio between the mass of
SOA formed and the mass of the reacted precursors (Stefenelli et al., 2019).

$$SOA\ yield\ (\%) = \frac{\sum_{i=1}^{n} LEVPi\_A}{(LEV\_G_0 + LEV\_A_0) - (LEV\_G + LEV\_A)} \tag{16}$$

where $LEVPi\_A$ represents a LEV oxidation product in the aerosol phase, subscript "0" refers to initial conditions and $n = 7$.
The terms represent mass concentrations. Formation of SOA from LEV degradation occurs rapidly (in the first 1-12 minutes
of the simulation), with maximum SOA yields ranging from 5 to 32% (Figure 4a). Among the simulated scenarios, the
largest SOA yields resulted from slightly faster heterogeneous chemistry scenarios (by one order of magnitude), initialized
with higher LEV_A concentrations (by one order of magnitude) and occurring on larger particles (larger diameter by a factor
of two) (Figure 4a and Table S1). The total aerosol mass (the sum of concentrations of all LEV-related aerosol species,
including the radicals) also increased by 14 % in the first minutes and remained constant throughout the simulation period.
This suggests that the multiphase chemistry of LEV along with its phase partitioning cannot be ignored in assessments of fire
air quality effects and can have variable effects on SOA yields depending on the initial conditions and aerosol properties.





### 3.3 Effects of LEV degradation on other gases

Implementation of LEV chemistry in models can also be used to consider its effects on other atmospheric species to better understand the effects of fire on air quality and atmospheric chemistry, such as the formation of tropospheric $O_3$ in the presence of $NO_x$ and VOC (both emitted from fires), conversion of $NO_x$ to other reactive nitrogen forms (including nitrated

LEV), interaction with key gas-phase species oxidants, etc. We studied effects in the model scenarios by comparing the concentrations of those key species obtained with LEV chemistry and those obtained without LEV chemistry (Figure 5 and Figure 6).

We found that LEV chemistry including G/P partitioning on average increases the concentrations of OH, nitrate radical ($NO_3$), $O_3$, nitric acid ($HNO_3$) and $NO_z$, while it decreases the concentrations of $N_2O_5$, $NO_x$ and total VOC (that does not

include LEV_G and LEV_A). These effects are the net result of full LEV chemistry in which species may be consumed or generated. For example, OH is consumed in reactions 1 and 10 but it is also generated directly in reactions 4 and 12, and indirectly through its precursor $HO_2$ that is generated by reactions 3, 6 and 12 (Table 1).

LEV chemistry modulates the concentration of reactive species that also interact with other VOC. Because LEV chemistry increases the concentrations of key oxidants (OH, $NO_3$, $O_3$), it causes the concentration of total VOC to decrease over time

due to increased availability of their oxidants. LEV chemistry also causes $NO_z$ to increase over time; this can mainly be explained by the formation of nitrated organic compounds (LEVP5_G and LEVP5_A) and $HNO_3$ in reactions 13 (Table 1) and 14 (Table 2). LEV chemistry also generates $NO_3$ precursors (such as $NO_2$) that may explain the net increase in $NO_3$ concentration (Figure 5).

We also studied the effects of LEV chemistry on the $O_3$ versus $NO_x$, $O_3$ versus VOC and $O_3$ versus VOC/$NO_x$ ratio

relationships as well as effects on the VOC/$NO_x$ ratio itself (Figure 7). While the decay of $NO_x$ slowed down the increase of $O_3$, the decay of VOC had no effect on the rate of $O_3$ formation when total VOC did not contain LEV_G and LEV_A. When the latter two were included in total VOC, the decay of total VOC also reduced the rate of the $O_3$ increase (linear slope of -0.2419 ± 0.001 ppb/ppbC) but not as much as $NO_x$ did (linear slope of -0.288 ± 0.008 ppb/ppb). The VOC/$NO_x$ ratio increases when LEV chemistry is considered, driving $O_3$ to reach higher concentrations (117 ppb) compared to the default

case (without LEV chemistry). Thus, when LEV chemistry operates in the system, the change in $O_3$ concentration is primarily driven by the change in $NO_x$ and only secondarily by the change in VOC.

### 3.4. Sensitivity analysis

Heterogeneous chemistry is the most sensitive aspect of the modelling approach in the present study. Here we assumed that the aerosol surface is composed of pure LEV and there are many factors that can interfere or inhibit heterogeneous chemistry

of a pure LEV substrate (section 2.3). These controls were lumped into a single factor (F) that we varied from a default case (1.0) to cases in which heterogeneous chemistry was up to three orders of magnitude slower. While available chamber experiments studies offered the opportunity to evaluate LEV degradation for a given heterogeneous reaction rate coefficient





that was reduced by certain F values (see section 3.1), other values of F are plausible. As a starting point, here we show how these F values influence the degradation time scale of LEV (Figure 8) and the SOA yields (Figure 4b). Within this wide

range of heterogeneous reactions rates (at constant $\alpha = 0.001$), the degradation time scale of LEV can be as long as 5 days in the gas phase and 7 days in the aerosol phase (when F = 0.001). These are larger time scales than those observed with reaction rates used in chamber comparisons (see section 3.1) and suggest that LEV can be transported and deposited both locally and regionally. Over these time scales, SOA yields vary within the same range (7-32%) as observed in the previous cases considered (see section 3.2). Compared to model scenarios evaluated by chamber experiments (Figure 4a), the SOA

yields start levelling out only after 4-5 days as opposed to 1 day, suggesting that some SOA still forms in the one to four-to-five day window.

We also tested the sensitivity of the mass accommodation coefficient ($\alpha$). Varying this by four orders of magnitude (0.001, 0.01, 0.1 and 1.0) showed little effect on LEV degradation (i.e., degradation was slightly faster when $\alpha = 1$) in comparison to the effect of slowing down the heterogeneous chemistry (F, as described above). The mass accommodation coefficient

appears to be more important when the G/P partitioning is modelled as gas-aerosol equilibrium reactions of which the partitioning coefficient is modelled with eq. 13. This is a different way to implement the G/P partitioning in the model, but it does not drive species phase transfer based on the theoretical surface equilibrium concentration (eq. 8 and 9).

## 4 Conclusions

Anhydrosugars emitted by biomass burning are key tracers of PyC and of carbon cycling throughout Earth system reservoirs.

However, relatively little is known about their degradation in any environment. A better understanding of the atmospheric degradation of anhydrosugars is necessary for both atmospheric and cryospheric sciences because it will improve the understanding of air quality effects of fire as well as the interpretation of levoglucosan records of fire, paleoclimate and paleovegetation recorded in the ice (Gambaro et al., 2008; Kawamura et al., 2012; Kehrwald et al., 2012; You and Xu, 2018). This study focused on the atmospheric degradation of anhydrosugars from the perspective of LEV, the most abundant

anhydrosugar emitted on a mass basis.

Using a 0-D modeling framework (BOXMOXv1.7), we implemented multiphase chemistry and G/P partitioning of LEV and its initial oxidation products (LEVCHEM_v1). We found that LEV degradation time scale ranges from 8-21 hours (aerosol-phase) to 1.5-3.5 days (gas-phase); however, model output was evaluated only for five hours through comparison to chamber measurements. In addition, we conducted a sensitivity analysis investigating a factor slowing down the heterogeneous

chemistry and found that longer degradation time scales may occur, ranging from 5 days (gas- phase) to 7 days (aerosol-phase). These longer time scales are similar to those of deposition (1-5 days) but are slightly shorter than that of regional transport (10 days), suggesting that both gas and aerosol LEV phases can be deposited locally but some fractions may be transported regionally. However, these time scales remain to be evaluated using more extensive measurements from chambers and fire plumes. Additional sensitivity analyses using larger initial aerosol LEV concentrations in chamber





simulations may result in longer degradation time scales of LEV aerosol concentration. Ultimately, implementation of the 0-D model development of this study into CTMs will help to clarify the regional transport and deposition of both LEV phases.

LEV degradation contributes to SOA formation that was quantified mainly through simulated SOA yields. Based on 5-h degradation time scales, simulated SOA yields ranged from 5 to 32 % and peaked in the first 1-12 minutes. Varying the heterogeneous chemistry rate by four orders of magnitude did not result in significantly different SOA yields (7-32%), but the decrease in the SOA yields was slower and extended to 4.5 days, consistent with the simulated degradation time scale of LEV in the same scenarios. The total PM mass (determined as the ratio of total aerosol concentration to initial LEV_A concentration) increased by 14% in the first 7 minutes of all simulations and remained essentially constant over time.

The addition of the multiphase LEV chemistry and the related G/P partitioning mechanism to the 0-D modelling framework has both direct and indirect effects on several gas-phase species. The average concentrations of OH, $NO_3$, $O_3$, $HNO_3$ and $NO_z$ increased, while those of $N_2O_5$, $NO_x$ and other VOC decreased. These changes are due to chemical reactions of the full LEV chemistry which simultaneously consume and generate reactive species. Other species, included in the total VOC, are indirectly influenced by the LEV chemistry via competition for oxidants or via the oxidant concentration mediated by LEV chemistry. The effects of LEV chemistry on $O_3$ are complex: while it slows down its rate of formation by modulating $NO_x$ and VOC concentrations, it increases the VOC/$NO_x$ ratio, which in turn leads to higher $O_3$ (117 ppb) compared to the case without LEV chemistry (90 ppb).

LEV chemistry facilitates the conversion of $NO_x$ to other reactive nitrogen forms (an increase of $NO_z$ versus time at an average $NO_z$ enhancement by 5 ppb). The effects of LEV chemistry on $NO_z$ occur directly through LEVP5, a nitrated organic degradation product, and indirectly via generation of $HNO_3$ or consumption of $N_2O_5$, NO and $NO_3$ in chemical reactions. LEV chemistry drives changes in major air pollutants making it unwise to ignore it in future assessments of fire effects on tropospheric $O_3$, nitrogen cycle (via $NO_z$) and carbon cycle (via VOC and aerosol-phase degradation products).

Future work should expand model development to include the degradation of the two LEV isomers (mannosan and galactosan) and to implement the full mechanism of anhydrosugar degradation into 3-D CTMs. The atmospheric implications of anhydrosugar degradation (i.e., SOA formation) and their tracing potential could then be evaluated more completely.

*Author contribution*. L. G. Suciu developed the LEVCHEM_v1model, ran CMAQ simulations to provide initial conditions for LEVCHEM_v1 simulations, ran simulations, gathered data from chamber experiments, analysed simulations results, evaluated model predictions, performed model sensitivity analysis, and wrote the manuscript. R. J. Griffin provided guidance for model development, evaluation, and sensitivity analysis, and critically reviewed the manuscript. C. A. Masiello provided critical review of the manuscript.

*Competing interests*. The authors declare that they have no conflict of interest.



*Code and data availability.* The model version that was used in this study (*BOXMOXv1.7*) to develop *LEVCHEM_v1* is

available at the following website: *https://boxmodeling.meteo.physik.uni-muenchen.de/* under the Christoph Knote (LMU Munich, Germany) / Jérôme Barré (ECMWF, UK) licence. The exact version of the *BOXMOXv1.7* model that was updated (*LEVCHEM_v1*) as well as the input data and scripts used to run numerical chamber simulations of which results are presented in this paper are archived on Zenodo (https://zenodo.org/record/3885786).

*Acknowledgements.* The authors wish to thank the BOXMOX developer, C. Knote, for the preliminary discussions regarding the development of a multiphase chemical mechanism. We are also grateful to Q. Z. Rasool and D. S. Cohan for sharing the data inputs needed for CMAQ simulations. Thanks to D. S. Cohan for useful discussions about model development and simulations. We thank the Department of Earth, Environmental and Planetary Sciences, Rice University, for providing the computational resources needed for the project and for IT support.

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



**Table 1 Homogeneous gas phase mechanism**

| Chemical reaction | Reference | Reaction rate coefficient [$cm^3\ molec^{-1}\ s^{-1}$] | Reference |
|---|---|---|---|
| 1. $LEV\_G + OH\ \{+ O_2\} \rightarrow LEVRO2\_G + H_2O$ | Bai et al. (2013); Jenkin et al. (1997) | $2.21 \times 10^{-12}$ | Bai et al. (2013) |
| 2. $LEVRO2\_G + NO \rightarrow LEVRO\_G + NO_2$ | Saunders et al. (2003) | $2.54 \times 10^{-12} \exp (360/T)$ | Saunders et al. (2003) |
| 3. $LEVRO\_G + O_2 \rightarrow LEVP1\_G + HO_2$ | Saunders et al. (2003) | $1.00 \times 10^{-14}$ | Seinfeld and Pandis (2006) |
| 4. $LEVRO2\_G + H_2O \rightarrow LEVP2\_G + OH + H_2O$ | Jenkin et al. (1997); Bai et al. (2013) | $1.00 \times 10^{-17}$ | Jenkin et al. (1997) |
| 5. $LEVP2\_G + LEV \rightarrow LEVP3\_G$ | Bai et al. (2013) | $3.10 \times 10^{-10} \exp (155/T)$ | Calculated in this study |
| 6. $LEVRO2\_G + M \rightarrow LEVP4\_G + HO_2 + M$ | Bai et al. (2013) | $5.76 \times 10^{-12} \exp (71/T)$ | Calculated in this study |
| 7. $2LEVRO2\_G \rightarrow 2LEVRO\_G + O_2$ | Saunders et al. (2003) | $2.70 \times 10^{-12}$ | Jenkin et al. (1997) |
| 8. $LEVRO2\_G + XO_2 \rightarrow LEVRO\_G + ROR + O_2$ | Saunders et al. (2003) | $2.70 \times 10^{-12}$ | Jenkin et al. (1997) |
| 9. $LEVRO2\_G + HO_2 \rightarrow LEVROOH\_G + O_2$ | Saunders et al. (2003) | $2.91 \times 10^{-13} \exp (1300/T)$ | Saunders et al. (2003) |
| 10. $LEVROOH\_G + OH \rightarrow LEVRO2\_G + H_2O$ | Emmons et al. (2010) | $3.80 \times 10^{-12} \exp (200/T)$ | Emmons et al. (2010) |
| 11. $LEV\_G + NO_3\ \{+ O_2\} \rightarrow LEVRO2\_G + HNO_3$ | Jenkin et al. (1997); Knopf et al. (2011) | $5.80 \times 10^{-16}$ | CB05TUCl_EPA (R77) |
| 12. $LEV\_G + O_3\ \{+ O_2\} \rightarrow LEVRO2\_G + O_2 + OH$ | Jenkin et al. (1997); Atkinson and Carter (1984) | $1.20 \times 10^{-14} \exp (2630/T)$ | CB05TUCl_EPA (R122) |
| 13. $LEV\_G + N_2O_5 \rightarrow LEVP5\_G + HNO_3$ | Gross et al. (2009) | $1.29 \times 10^{-14}$ | Calculated this study |

{} Species concentration not included in the reaction rate (i.e., reaction of LEV_G radical with $O_2$ is assumed to be instantaneous)

**Table 2 Heterogeneous mechanism**

| Chemical reaction | Reference | Uptake coefficient[a] | Reference |
|---|---|---|---|
| 1. $LEV\_A\ \{+ OH\} \rightarrow LEVR\_A + H_2O$ | Bai et al. (2013); Jenkin et al. (1997) | 0.91 | Kessler et al. (2010) |
| 2. $LEVR\_A\ \{+ O_2\} \rightarrow LEVRO2\_A$ | Saunders et al. (2003) | 0.41 | Calculated this study |
| 3. $LEVRO2\_A\ \{+ NO\} \rightarrow LEVRO\_A + NO_2$ | Saunders et al. (2003) | 0.36 | Calculated this study |
| 4. $LEVRO\_A\ \{+O_2\} \rightarrow LEVP1\_A + O_2$ | Saunders et al. (2003) | 0.41 | Calculated this study |
| 5. $LEVRO2\_A\ \{+ H_2O\} \rightarrow LEVP2\_A + OH + H_2O$ | Bai et al. (2013); Jenkin et al. (1997) | 0.22 | Calculated this study |
| 6. $LEVP2\_A\ \{+ LEV\_G\} \rightarrow LEVP3\_A$ | Bai et al. (2013) | 0.92 | Calculated this study |
| 7. $LEVRO2\_A\ \{+ LEVRO2\_G\} \rightarrow LEVRO\_A + LEVRO\_G + O_2$ | Saunders et al. (2003) | 0.85 | Calculated this study |
| 8. $LEVRO2\_A\ \{+ HO_2\} \rightarrow LEVROOH\_A + O_2$ | Saunders et al. (2003) | 0.33 | Calculated this study |
| 9. $LEVROOH\_A\ \{+ OH\} \rightarrow LEVRO2\_A + H_2O$ | Emmons et al. (2010) | 0.27 | Calculated this study |
| 10. $LEV\_A\ \{+ OH\} \rightarrow LEVP6\_A + LEVR1\_A + H_2O$ | Kessler et al. (2010) | 0.27 | Calculated this study |
| 11. $LEVR1\_A\ \{+O_2\} \rightarrow LEVP7\_A + HO_2$ | Saunders et al. (2003) | 0.41 | Calculated this study |
| 12. $LEV\_A\ \{+ NO_3\} \rightarrow LEVR\_A + HNO_3$ | Jenkin et al. (1997); Knopf et al. (2011) | 1.29 | Knopf et al. (2011) |
| 13. $LEV\_A\ \{+ O_3\} \rightarrow LEVR\_A + O_2 + OH$ | Jenkin et al. (1997); Atkinson and Carter (1984) | 0.013 | Knopf et al. (2011) |
| 14. $LEV\_A\ \{+ N_2O_5\} \rightarrow LEVP5\_A + HNO_3$ | Gross et al. (2009) | 0.027 | Knopf et al. (2011) |

[a]The uptake coefficient used in the calculation of the heterogeneous reaction rate coefficient (see eq. 5)

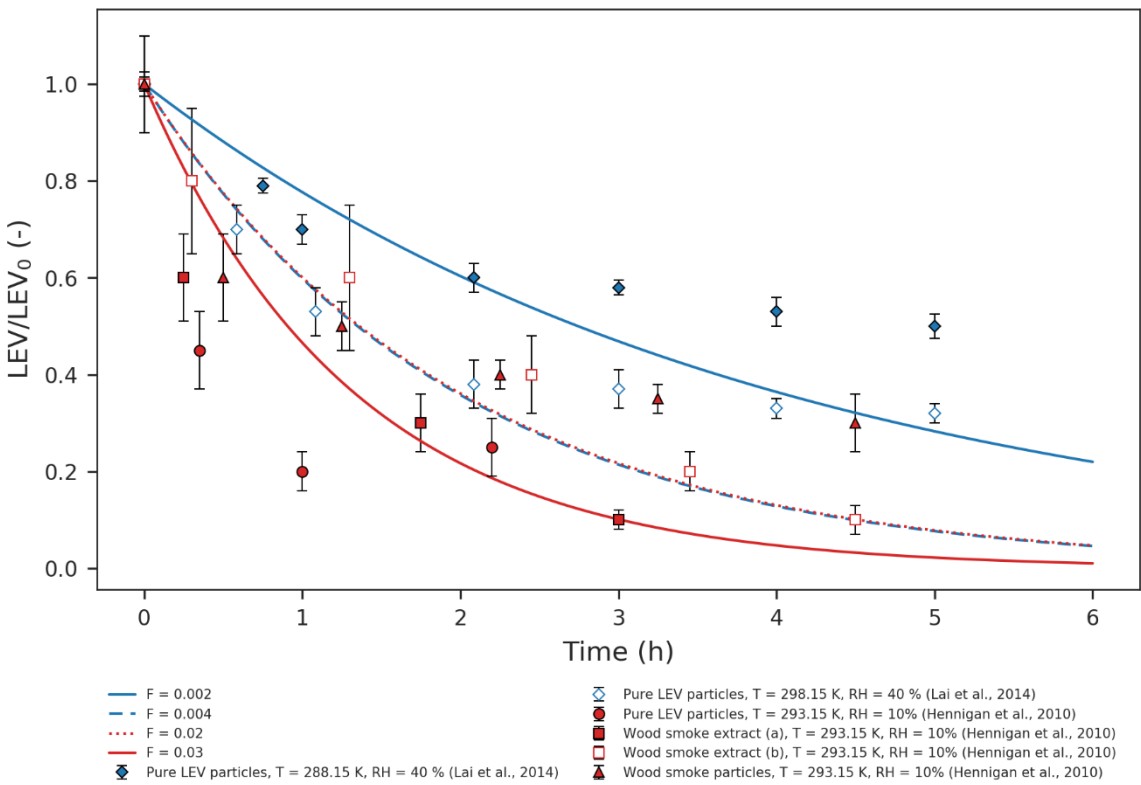


**Figure 1 Simulated LEV degradation (lines) and measured LEV degradation (points); Color represents conditions from different chamber experiments taken from two studies (red – Hennigan et al. (2010) and blue – Lai et al. (2014) used in the simulations. LEV concentration is normalized by the initial concentration (LEV/LEV$_0$).**





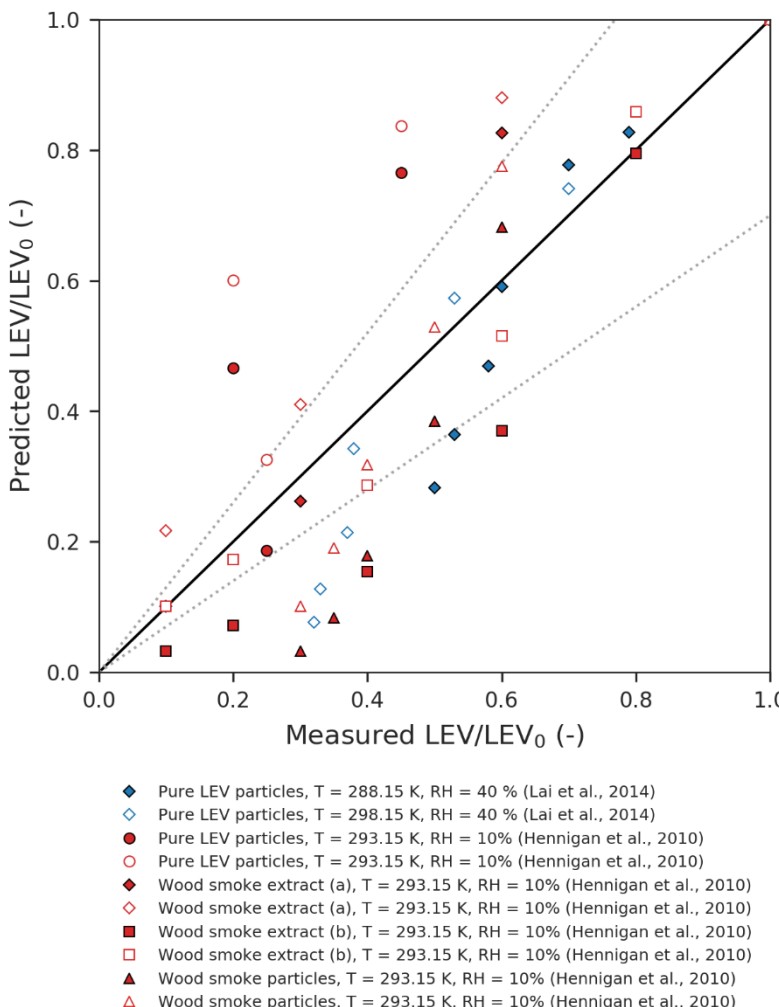


**Figure 2 Parity plot of predicted versus measured LEV concentration (normalized by the initial concentration). The dotted lines represent the ± 30% error margins.**



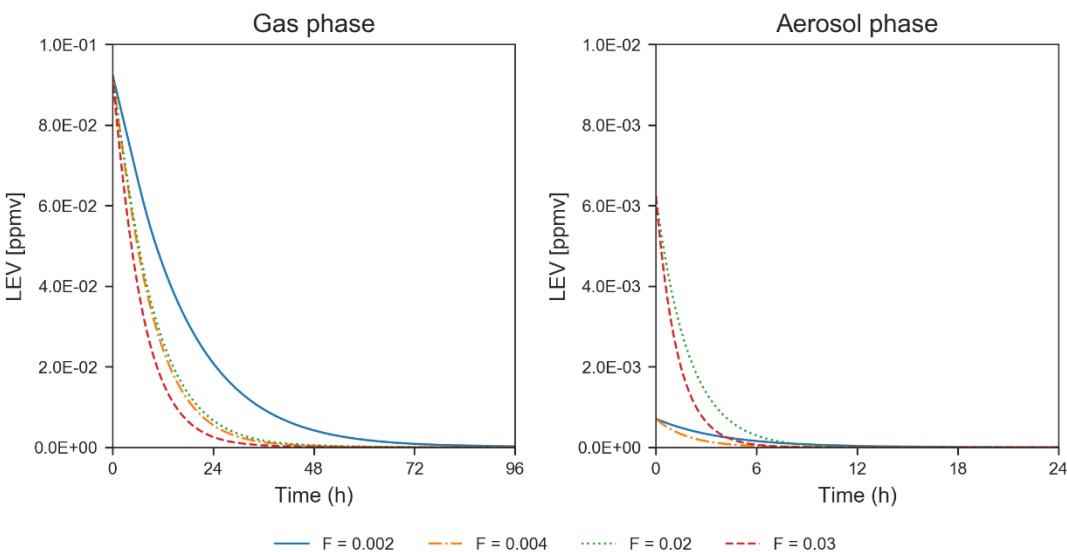

**Figure 3** Degradation of LEV in simulations at constant mass accommodation coefficient (α = 0.001) (conditions from Hennigan et al. (2010) when F=0.02 and 0.03, and from Lai et al. (2014) when F=0.002 and 0.004). Note the change in the scale of the axes between the two panels.

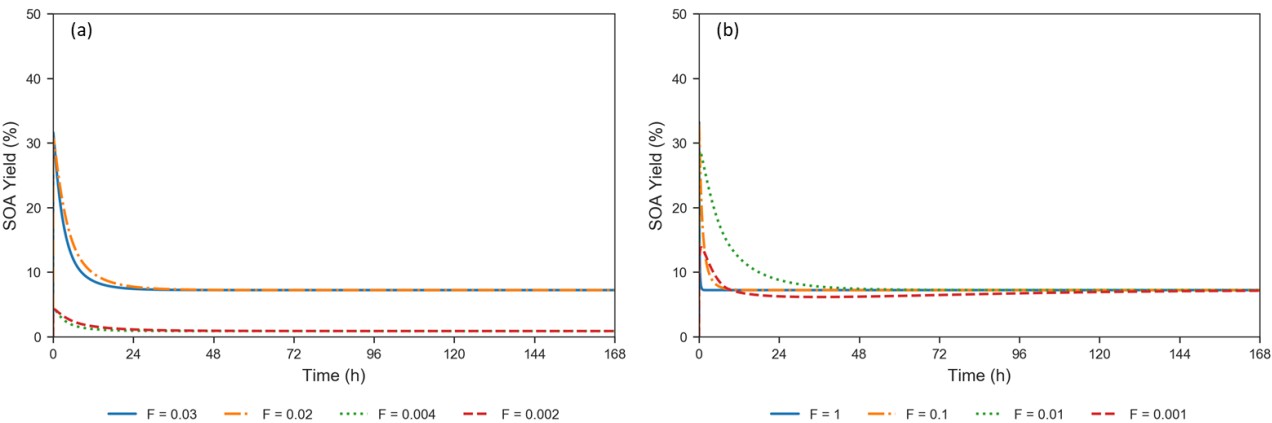

**Figure 4 (a)** Evolution of SOA yields from LEV degradation using valid simulations at constant mass accommodation coefficient (α = 0.001) (conditions from Hennigan et al. (2010) when F=0.02 and 0.03, and from Lai et al. (2014) when F=0.002 and 0.004). **(b)** Effect of varying the heterogeneous reaction rate coefficient by 4 orders of magnitude, at constant mass accommodation coefficient (α = 0.001) (conditions from Hennigan et al. (2010)).



**Figure 5 Effects of LEV chemistry on OH, NO₃, O₃ and NOₓ (in red, relative to the case without LEV chemistry shown in black or grey). The time series represent averages of simulations performed with LEV chemistry (dashed red line) and without LEV chemistry (black line) over the 5-h time scale. The box plots show the distributions of the species concentration for the entire 5 hours. Note that findings shown here are determined over a range of F values depending on experimental conditions.**

**Figure 6 Effects of LEV chemistry on HNO₃, N₂O₅, NOz and VOC (in red, relative to the case without LEV chemistry shown in black or grey). The time series represent averages of simulations performed with LEV chemistry (dashed red line) and without LEV chemistry (black line) over the 5-h time scale. The box plots show the distributions of the species concentration for the entire 5 hours. Note that findings shown here are determined over a range of F values depending on experimental conditions.**



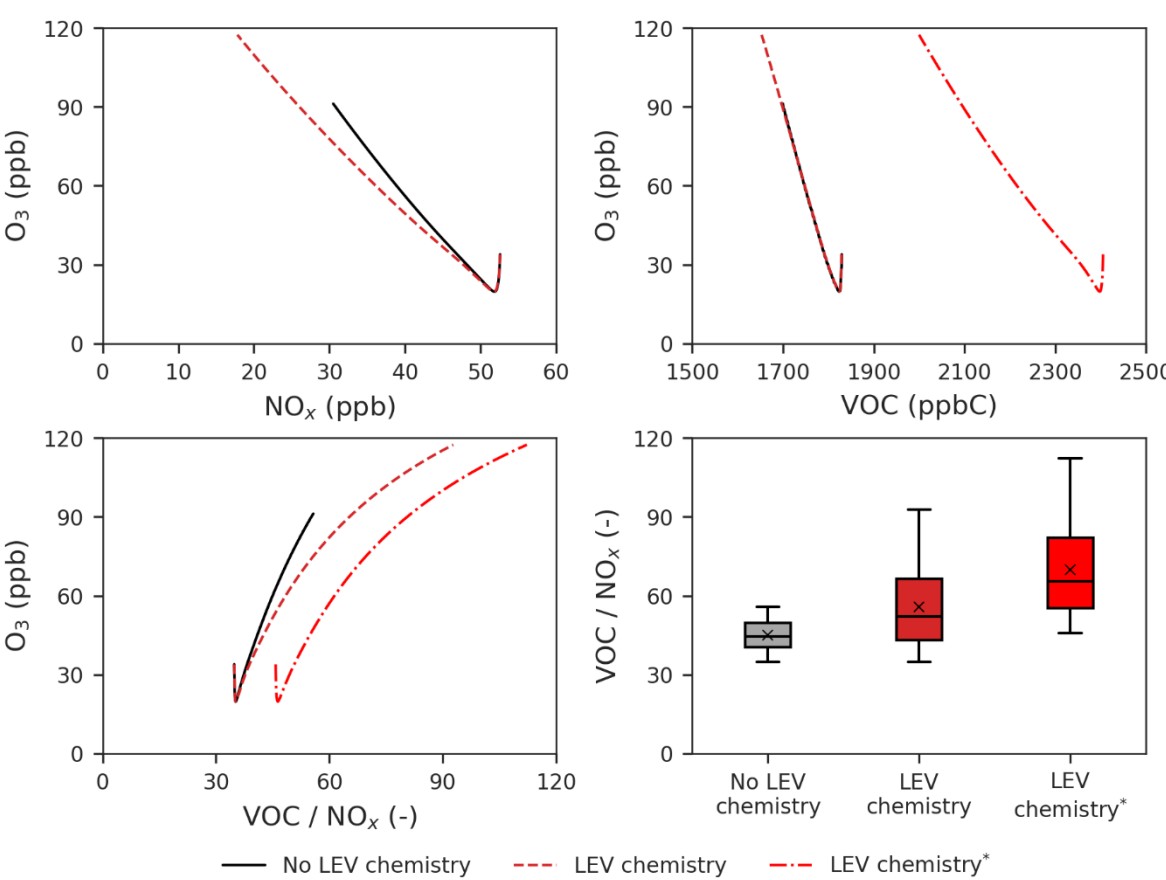

**Figure 7 Effects of LEV chemistry on the $O_3$ versus $NO_x$, $O_3$ versus VOC and $O_3$ versus VOC/$NO_x$ ratio relationships and on the VOC/$NO_x$ ratio. The two cases in red (with LEV chemistry) refer to the two ways in which VOC was determined (with/without LEV_G and LEV_A). The asterisk refers to the inclusion of LEV_G and LEV_A in the total VOC. All the plots show simulation results at the 5-h time scale. Note that findings shown here are determined over a range of F values depending on experimental conditions.**



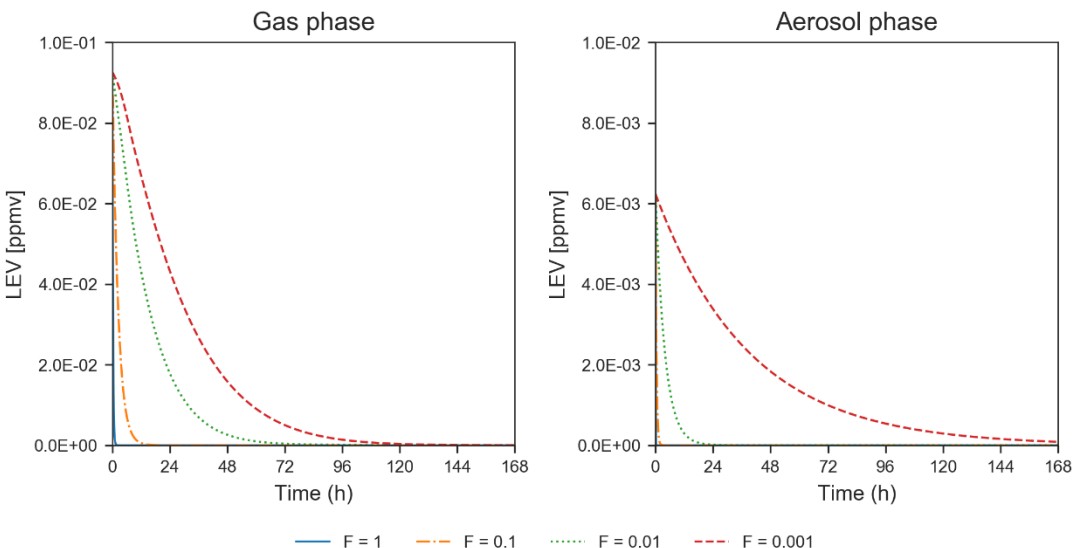

**Figure 8 Degradation of LEV by varying the heterogeneous reaction rate coefficient by 4 orders of magnitude, at constant mass accommodation coefficient ($\alpha = 0.001$) (conditions from Hennigan et al. (2010)). Note that the y-axis scale changes between the concentrations presented for the two phases.**

510