# Peer review of "A zero-dimensional view of atmospheric degradation of levoglucosan (LEVCHEM\_v1) using numerical chamber simulations"

_Geoscientific Model Development, 2020_

## Referee Comment (RC1) · Andrew May (Referee) · 29 Jul 2020

**Review of gmd-2020-189**
Suciu et al., "A zero-dimensional view of atmospheric degradation of levoglucosan using numerical chamber simulations"

**Summary:**
The authors presents a numerical study investigating the atmospheric degradation of levoglucosan, which itself isn't necessarily new, but adding the explicit chemical mechanisms and the effect of levoglucosan chemistry on trace gases is not something that I've seen previously, so those are nice additions. Arguably, this is a modeling framework that can be integrated directly into a chemical transport model, so it has substantial potential for utility in the modeling world. I don't have a large number of comments, but I do have one that is fairly significant (Major Comment), one related comment (General Comment), three others that are more towards clarification (Specific Comments), and one suggestion.

Overall, this is a good manuscript, and it will be valuable for the community. Therefore, my recommendation is to reconsider after at least the Major Comment and Specific Comments have been addressed (the authors can take or leave my suggestion).

I have never "signed" a review previously, but I felt that it was necessary here due to the relationship of this manuscript with my previous work. I will also add that I am happy to continue this discussion, as needed, if the authors require any clarification. I am uncertain of the protocol here, but I imagine that we could have an "offline" discussion over email that could ultimately be included as part of the Interactive Discussion.

**Major Comment:**
- There appears to be a fundamental error regarding the volatility ($C_i^*$) of levoglucosan in this modeling framework.
  - Table S1 shows that $C_i^* = 3 \times 10^{-3}$ kg m$^{-3}$ = $3 \times 10^6$ μg m$^{-3}$, which would imply that that vast majority of the levoglucosan should be in the gas phase (e.g., if $C_{OA} = 30$ μg m$^{-3}$, 99.999% would be in the gas phase). This does not match with the initial distribution of the gas-phase and aerosol-phase concentration in Table S1.
  - Experimentally, we showed that this was more like 13 μg m$^{-3}$ in May et al. (2012).
    - We speculated that this was why Hennigan et al. (2010) observed uptake coefficients ($\gamma$) greater than 1, and this could be a plausible explanation for the results in Knopf et al. (2011) for NO$_3$ radicals.
  - Moreover, a structure-activity relationship (Mansouri et al., 2018) predicts a vapor pressure that yields $C_i^* \approx 1$ μg m$^{-3}$.
  - Perhaps this issue with $C_i^*$ is the reason why $\alpha = 0.0001$ is required to demonstrate good agreement in Figure 1?

**General Comment:**
- Related to the Major Comment, Kulmala and Wagner (2001) provide a theoretical relationship between mass accommodation coefficients ($\alpha$) and $\gamma$. One key point that they make is that $\alpha \geq \gamma$ (depending on the Knudsen number). In this work, $\gamma \gg \alpha$, based on Table 2. Therefore, the authors should either constrain $\alpha > 0.1$ or provide some justification for $\alpha \ll \gamma$.
  - I suspect that this issue may work itself out once the Major Comment has been resolved, but it is worth noting in the event that it does not.

**Specific Comments:**
- Lines 198-203: As I'm reflecting on this after having read the full draft, it seems like these "F" values could be related to the mass fraction of levoglucosan in the particles. For example, F = 0.03 is the slope

of "$m/z$ 60" (proxy for anhydrosugars in the Aerosol Mass Spectrometer world) vs. OA concentration in Figure 7 in Sullivan et al. (2014). I haven't carefully read through Lai et al. (2014), but conceivably, they could have used lower mass-mixing ratios of levoglucosan in their particles. I bring this up because as written, F essentially sounds like a "fudge factor" to get the model to agree with observations. Maybe something like "We expect F to be, at a maximum, 0.1 due to observed mass fractions in biomass burning organic aerosol, and this value may decrease due to different levoglucosan sources or experimental conditions"

- Lines 203-204: I have no recollection of making this claim regarding $\alpha$ in May et al. (2013), and quickly reviewing that paper, it appears that the closest thing to that is that there appeared to be no mass transfer limitations for the evaporation of biomass burning smoke. In May et al. (2012), we did include some discussion of this in the online supporting information, but that was more focused on the fact that even if we decrease $\alpha < 1$, the evaporation timescale is shorter than heterogeneous chemistry timescale. I guess that my point here is that my previous work has been mis-interpreted.

  These results, coupled with $\gamma \approx 1$ (presumably, based on Kessler et al. (2010)?) and Kulmala and Wagner (2001) suggesting that $\alpha \geq \gamma$, seem to imply that $\alpha \approx 1$.

- Section 3.2 and Figure 4: I am a bit confused by this discussion about yield, which is perhaps based on an *a priori* expectation that SOA yield increases with increased photochemical aging. Is the implication that initially, many of the oxidation products remain in the particle phase (t < ~ 6 hr) but as the chemistry continues, heterogeneous oxidation results in more volatile products until a steady-state is reached (at t ~ 24 -36 hrs?

**Suggestion**
- Another dataset to consider would be that from Pratap et al. (2019). They also modeled their own data, explicitly accounting for gas-particle partitioning and chamber wall loss in addition to oxidation chemistry, in predicting particle-phase levoglucosan concentrations. If nothing else, these data can provide another valuable test set at colder temperatures than Hennigan et al. (2010) and Lai et al. (2014).

**References**
Hennigan, C. J., Sullivan, A. P., Collett, J. L. and Robinson, A. L.: Levoglucosan stability in biomass burning particles exposed to hydroxyl radicals, Geophys. Res. Lett., 37(9), L09806, doi:10.1029/2010GL043088, 2010.
Kessler, S. H., Smith, J. D., Che, D. L., Worsnop, D. R., Wilson, K. R. and Kroll, J. H.: Chemical sinks of organic aerosol: kinetics and products of the heterogeneous oxidation of erythritol and levoglucosan., Environ. Sci. Technol., 44(18), 7005–10, doi:10.1021/es101465m, 2010.
Knopf, D. A., Forrester, S. M. and Slade, J. H.: Heterogeneous oxidation kinetics of organic biomass burning aerosol surrogates by O3, NO2, N2O5, and NO3, Phys. Chem. Chem. Phys., 13(47), 21050, doi:10.1039/c1cp22478f, 2011.
Kulmala, M. and Wagner, P. E.: Mass accommodation and uptake coefficients — a quantitative comparison, J. Aerosol Sci., 32(7), 833–841, doi:10.1016/S0021-8502(00)00116-6, 2001.
Lai, C., Liu, Y., Ma, J., Ma, Q. and He, H.: Degradation kinetics of levoglucosan initiated by hydroxyl radical under different environmental conditions, Atmos. Environ., 91, 32–39, doi:10.1016/j.atmosenv.2014.03.054, 2014.
Mansouri, K., Grulke, C. M., Judson, R. S. and Williams, A. J.: OPERA models for predicting physicochemical properties and environmental fate endpoints, J. Cheminform., 10(1), 10, doi:10.1186/s13321-018-0263-1, 2018.
May, A. A., Levin, E. J. T., Hennigan, C. J., Riipinen, I., Lee, T., Collett, J. L., Jimenez, J. L., Kreidenweis, S. M. and Robinson, A. L.: Gas-particle partitioning of primary organic aerosol emissions:

3. Biomass burning, J. Geophys. Res. Atmos., 118(19), 11,327-11,338, doi:10.1002/jgrd.50828, 2013.

May, A. A., Saleh, R., Hennigan, C. J., Donahue, N. M. and Robinson, A. L.: Volatility of organic molecular markers used for source apportionment analysis: measurements and implications for atmospheric lifetime., Environ. Sci. Technol., 46(22), 12435–44, doi:10.1021/es302276t, 2012.

Pratap, V., Bian, Q., Kiran, S. A., Hopke, P. K., Pierce, J. R. and Nakao, S.: Investigation of levoglucosan decay in wood smoke smog-chamber experiments: The importance of aerosol loading, temperature, and vapor wall losses in interpreting results, Atmos. Environ., 199, 224–232, doi:10.1016/j.atmosenv.2018.11.020, 2019.

Sullivan, A. P., May, A. A., Lee, T., McMeeking, G. R., Kreidenweis, S. M., Akagi, S. K., Yokelson, R. J., Urbanski, S. P. and Collett Jr., J. L.: Airborne characterization of smoke marker ratios from prescribed burning, Atmos. Chem. Phys., 14(19), 10535–10545, doi:10.5194/acp-14-10535-2014, 2014.

---

## Referee Comment (RC2) · Anonymous Referee #2 · 21 Aug 2020

This paper describes a 0-D modeling framework that has been developed to examine the atmospheric degradation of levoglucosan, an important biomass burning tracer. The model employs the BOXMOX model and is called LEVCHEM_v1. The authors have updated the chemical mechanisms to include degradation of levoglucosan and intermediary products as well as added a gas-particle partitioning mechanism to account for effects of evaporation and condensation on the concentration of levoglucosan. Additionally, they have run their model on data from photo-oxidation chamber experiments of levoglucosan. The results are presented and discussed.

Overall, this is a very good paper. The authors have done a great job explaining their

model. The paper seems scientifically sound and is easy to follow. I would highly encourage the authors to try out their model on ambient levoglucosan data from aircraft campaigns that were able to follow smoke plumes over time such as WINTER and WE-CAN. I really just have a handful of minor comments outlined below that need to be addressed before the paper can be considered for publication.

General Comments: In the paper it seems to go back and forth if it is written as gas-phase and aerosol-phase (with hyphens) or gas phase and aerosol phase (without hyphens). This should be checked throughout the entire text.

Specific Comments: Line 73 – Suggest adding an of after developed use

Line 173 – Suggest changing particle phase to particle-phase

Line 260 – The comma after Supplemental Information) should be a period

Line 340 – There is an extra space in gas-phase

Line 393 – In the Alvardo et al. reference, periods are missing in the initials for author Akagi

Line 394 – In the Arangio et al. reference, believe oh should be capitalized

Figure 3 -In the plot titles, the hyphen is missing in Gas-phase and Aerosol-phase

Figure 8 -In the plot titles, the hyphen is missing in Gas-phase and Aerosol-phase

Figures S3, S4, S5, S6 -In caption, the hyphen is missing in gas-phase and aerosol-phase

---

## Author Comment (AC1) · 13 Oct 2020

**Author response to reviewers of the gmd-2020-189 manuscript**

We thank both reviewers for their constructive comments and helpful suggestions to improve our manuscript. Our responses to their comments are provided below, separately for each reviewer (sections A and B). At the end, we provide a third section (C), describing additional changes resulting from our answers to the reviewers' comments.

**A. Reviewer 1 (Andrew May)**

Major comment

*Reviewer comment:*

"There appears to be a fundamental error regarding the volatility ($C_i^*$) of levoglucosan in this modeling framework. Table S1 shows that $C_i^* = 3 \times 10^{-3}$ kg m$^{-3}$ = 3 x 10$^6$ µg m$^{-3}$, which would imply that that vast majority of the levoglucosan should be in the gas phase (e.g., if COA = 30 µg m$^{-3}$, 99.999% would be in the gas phase). This does not match with the initial distribution of the gas-phase and aerosol-phase concentration in Table S1. Experimentally, we showed that this was more like 13 µg m$^{-3}$ in May et al. (2012). We speculated that this was why Hennigan et al. (2010) observed uptake coefficients ($\gamma$) greater than 1, and this could be a plausible explanation for the results in Knopf et al. (2011) for NO$_3$ radicals. Moreover, a structure-activity relationship (Mansouri et al., 2018) predicts a vapor pressure that yields $C_i^* \approx 1$ µg m$^{-3}$. Perhaps this issue with $C_i^*$ is the reason why $\alpha = 0.0001$ is required to demonstrate good agreement in Figure 1?"

*Author comment:*

Indeed, the $C_i^*$ value we used in our model is likely too large and we thank the reviewer for pointing this out. In our model we used the maximum value of a range from May et al. (2013). However, levoglucosan may have a $C_i^*$ value closer to the lower end of that range, since it is not very volatile.

To address this, we ran new simulations by considering both suggestions ($C_i^* = 13$ µg m$^{-3}$ and $C_i^* = 1$ µg m$^{-3}$). We confirm that the model also predicts well the LEV degradation at larger $\alpha$ values than 0.001, respectively 0.01 and 0.1 (using $C_i^* = 13$ µg m$^{-3}$) and 1.0 (using $C_i^* = 1$ µg m$^{-3}$). The variations in the $\alpha$ response to $C_i^*$ are driven by the different experimental conditions used to initialize the model. For example, using conditions from Hennigan et al. (2010), the model performed well at $\alpha = 0.1$ ($C_i^* = 13$ µg m$^{-3}$) and at $\alpha = 1.0$ ($C_i^* = 1$ µg m$^{-3}$). Conditions from Lai et al. (2014) worked well only at $\alpha = 0.01$ ($C_i^* = 13$ µg m$^{-3}$). However, we chose to report results only for cases when $C_i^* = 13$ µg m$^{-3}$ was used because this value worked well with respect to $\alpha$, model evaluation and model performance, for all the modeled scenarios, including the conditions from an additional experimental study that the reviewer has suggested (Pratap et al., 2019). These results are shown in revised Figures 1 and 2 below. Model performance was not negatively affected; on the contrary, the accuracy improved slightly (from 48% to 47%).

[Figure]

**Figure 1 Simulated LEV degradation (lines) and measured LEV degradation (points); Color represents conditions from different chamber experiments taken from three studies (red – Hennigan et al. (2010), blue – Lai et al. (2014) and green – Pratap et al. (2019)) used in the simulations. LEV concentration is normalized by the initial concentration (LEV/LEV$_0$).**

[Figure]

**Figure 2 Parity plot of predicted versus measured LEV concentration (normalized by the initial concentration). The dotted lines represent the ± 30% error margins.**

General comment

*Reviewer comment:*

"Related to the Major Comment, Kulmala and Wagner (2001) provide a theoretical relationship between mass accommodation coefficients ($\alpha$) and $\gamma$. One key point that they make is that $\alpha \geq \gamma$ (depending on the Knudsen number). In this work, $\gamma \gg \alpha$, based on Table 2. Therefore, the authors should either constrain $\alpha > 0.1$ or provide some justification for $\alpha \ll \gamma$. I suspect that this issue may work itself out once the Major Comment has been resolved, but it is worth noting in the event that it does not."

*Author comment:*

We confirm that by correcting the $C_i^*$ used in our model, we obtained good model predictions at $\alpha$ values that are 1-2 orders of magnitude larger compared to previous simulations at much larger $C_i^*$ values (see our response to the major comment above). These $\alpha$ values (0.01 and 0.1) are still smaller than those of the uptake coefficients ($\gamma$) for most of the chemical species, including for levoglucosan. However, as seen in Table 2, the great majority of the $\gamma$ values were computed in this study, in the absence of their experimental measurements. In eq. 7, we assume a similar 2$^{nd}$ order heterogeneous reaction rate for all the species (see line 165); this may bias our calculations of $\gamma$ towards larger values. For $\gamma$ values in the order of $10^{-1}$ (OH uptake by levoglucosan, for example) and Knudsen numbers in the order of $10^{-1}$ (all modeled cases), the corresponding $\alpha$ should be 0.1, according to Fig. 1 in Kulmala and Wagner (2001). This is true when we model conditions from Hennigan et al. (2010); thus, in this case the $\alpha \geq \gamma$ criterion is satisfied. Modeled conditions from Lai et al. (2014) and Pratap et al. (2019) do not meet this criterion for levoglucosan because, for similar Kn and $\gamma$ values, the model worked well (compared to experimental data) only at $\alpha = 0.01$; in these cases, $\alpha < \gamma$. However, for other species with smaller $\gamma$ ($O_3$ and $N_2O_5$), all the modeled cases in our study satisfy the criterion $\alpha \geq \gamma$. It is worth noting here that the effective $\alpha$ values we found in our study, by comparing model predictions with data, have inherent uncertainties associated with both the data and the model.

To address this reviewer's comment directly in the manuscript:

(*i*) We introduce the relationship between $\alpha$ and $\gamma$ described by Kulmala and Wagner (2001) by rephrasing the sentence on line 205:

> "*The mass accommodation coefficient is related to the G/P partitioning mechanism (eq. 14) and the uptake coefficient ($\gamma$). Theoretically, $\alpha \geq \gamma$, depending on the Knudsen number (Kulmala and Wagner, 2001).*"

*ii*) We discuss our results with respect to $\alpha$ and $\gamma$ by adding the following paragraph starting on line 231:

> "*These $\alpha$ values are smaller than those of $\gamma$ for most of the chemical species, including for levoglucosan. However, as seen in Table 2, the great majority of the $\gamma$ values were computed in this study, in the absence of their experimental measurements. In eq. 7, we assume a similar 2$^{nd}$ order heterogeneous reaction rate for all the species; this may bias our calculations of $\gamma$ towards larger values. For $\gamma$ values in the order of $10^{-1}$ (OH uptake by levoglucosan, for example) and Knudsen numbers in the order of $10^{-1}$ (all modeled cases), the corresponding $\alpha$ should be ~0.1, according to Fig. 1 in Kulmala and Wagner (2001). This is true when we model conditions from Hennigan et al. (2010); thus, in this case the $\alpha \geq \gamma$ criterion is marginally satisfied. Modeled conditions from Lai et al. (2014) and Pratap et al. (2019) do not meet this criterion for levoglucosan because, for similar Kn and $\gamma$ values, the model worked well (compared to experimental data) only at $\alpha = 0.01$; in these cases, $\alpha < \gamma$. However, for other species with smaller $\gamma$ ($O_3$ and $N_2O_5$), all the modeled cases in our study satisfy the criterion $\alpha \geq \gamma$. It is worth noting here that the effective $\alpha$ values we found in our study by comparing model predictions with data have inherent uncertainties associated with both the data and the model.*"

Specific comments

*Reviewer comment:*

"Lines 198-203: As I'm reflecting on this after having read the full draft, it seems like these "F" values could be related to the mass fraction of levoglucosan in the particles. For example, F = 0.03 is the slope of "$m/z$ 60" (proxy for anhydrosugars in the Aerosol Mass Spectrometer world) vs. OA concentration in Figure 7 in Sullivan et al. (2014). I haven't carefully read through Lai et al. (2014), but conceivably, they could have used lower mass-mixing ratios of levoglucosan in their particles. I bring this up because as written, F essentially sounds like a "fudge factor"

to get the model to agree with observations. Maybe something like "We expect F to be, at a maximum, 0.1 due to observed mass fractions in biomass burning organic aerosol"

*Author comment:*

This is an interesting suggestion. We used F precisely for accounting for the fact that there are many factors that may inhibit heterogeneous chemistry while the heterogeneous reaction rate ($k_{SFC\_REACTION}$) does not account for that. Model evaluation confirms that by not constraining $k_{SFC\_REACTION}$ by F, results are not realistic. We see that F could be a mass fraction accounting for levoglucosan (or other particle-phase species) that is available to react with gas-phase oxidants. At a minimum, we found F to be 0.001 (conditions from Pratap et al., 2019), while at a maximum, F was 0.02-0.03 (conditions from Hennigan et al., 2010) (see Figure 1 above). We also think that these variations in F are driven by different temperature (and possibly relative humidity) conditions modeled, with higher temperature sustaining larger F and lower temperature supporting lower F. Therefore, we added your suggestion starting on line 203 as follows:

> *"We expect F to be, at a maximum, 0.1 due to observed mass fractions in biomass burning organic aerosols (Sullivan et al., 2014)."*

*Reviewer comment:*

"Lines 203-204: I have no recollection of making this claim regarding α in May et al. (2013), and quickly reviewing that paper, it appears that the closest thing to that is that there appeared to be no mass transfer limitations for the evaporation of biomass burning smoke. In May et al. (2012), we did include some discussion of this in the online supporting information, but that was more focused on the fact that even if we decrease α < 1, the evaporation timescale is shorter than heterogeneous chemistry timescale. I guess that my point here is that my previous work has been mis-interpreted.

These results, coupled with γ ≈ 1 (presumably, based on Kessler et al. (2010)?) and Kulmala and Wagner (2001) suggesting that α ≥ γ, seem to imply that α ≈ 1."

*Author comment:*

On Line 204, we refer to α as "…(which is 0.1 for a system in equilibrium (May et al., 2013)…". We inferred this from paragraph 18/page 11,330 (May et al., 2013), where α ≥ 0.1 is considered a good value for a system in equilibrium, as has been found in different chamber systems. We rephrased the sentence starting on line 203 as follows:

> *"In addition, we varied the mass accommodation coefficient (see eq. 14) from a default case of 0.1 (which is the lower limit of α for a system in equilibrium (May et al., 2013)) to lower values (0.01 and 0.001) and larger values (1.0). It was necessary to vary α since its value is unknown for levoglucosan and its degradation products."*

*Reviewer comment:*

"Section 3.2 and Figure 4: I am a bit confused by this discussion about yield, which is perhaps based on an *a priori* expectation that SOA yield increases with increased photochemical aging. Is the implication that initially, many of the oxidation products remain in the particle phase (t < ~ 6 hr) but as the chemistry continues, heterogeneous oxidation results in more volatile products until a steady-state is reached (at t ~ 24 -36 hrs?)"

*Author comment:*

If we look at Figures S3-S7 (see section C/Supplemental Information), about the same number of products (~3) is present in both phases, but the products differ between the two phases. Heterogeneous chemistry is the driver of these yields in the first hours. Condensation is negligible, since LEVP4, which is a product of gas-phase chemistry only, is not present in the aerosol phase; it only appears in the aerosol phase at the end of the simulation, in some cases (Fig. S4-S7). Evaporation is not important either since the products LEVP6 and LEVP7, generated by heterogeneous chemistry only, are not seen in the gas phase at all. Most of the oxidation products in the aerosol phase remain in the aerosol phase over the entire simulation period, except for LEVP5 and LEVP1 that may partition to the gas phase after 6 hours. The steady-state is reached because the concentration of precursors is nearzero and the oxidation products from heterogeneous chemistry and condensation (i.e., LEVP4 only) remain in the aerosol phase. So, yes, your interpretation is correct. Therefore, we supplemented this section by adding the following text starting on line 274 (for updated Figure 4, see section C/Manuscript):

> *"These high SOA yields in the first 6 hours are the result of rapid conversion of the precursors to aerosol-phase products, mainly due to heterogeneous chemistry. Because these products are not seen in the gas phase, evaporation does not influence the SOA yields in this early stage of the simulation; condensation of gas-phase products (LEVP4 and LEVP5) is also negligible (see Fig. S3-S7 in Supplemental Information). Most of the oxidation products remain in the aerosol phase over the entire simulation period, except for LEVP5 and LEVP1 that may partition to the gas phase. SOA yield reaches steady-state at ~24-26 hours due to near-zero concentrations of the two precursors and the presence of oxidation products from heterogeneous chemistry and G/P partitioning (i.e., condensation of LEVP4) in the aerosol phase."*

Suggestion

*Reviewer comment:*

"Another dataset to consider would be that from Pratap et al. (2019). They also modeled their own data, explicitly accounting for gas-particle partitioning and chamber wall loss in addition to oxidation chemistry, in predicting particle-phase levoglucosan concentrations. If nothing else, these data can provide another valuable test set at colder temperatures than Hennigan et al. (2010) and Lai et al. (2014)."

*Author comment:*

We thank the reviewer for this suggestion. We considered it in our simulations (see revised Figs. 1 and 2 above). Because conditions of this study represent wintertime conditions (T = 10°C/283.15 K), the model line falls in the proximity of the model conditions from Lai et al. (2014) (T = 15°C/288.15 K) and works for an F of similar magnitude ($10^{-3}$) and α (0.01). We also refer to this study in the manuscript and Supplemental Information (see section C for additional changes). Simulation results based on this study are also included in Figs. 3 and 4 (see section C/Manuscript) and Fig. S7 (see section C/Supplemental Information).

**B. Reviewer 2 (anonymous)**

Suggestion

*Reviewer comment:*

"I would highly encourage the authors to try out their model on ambient levoglucosan data from aircraft campaigns that were able to follow smoke plumes over time such as WINTER and WECAN."

*Author comment*: This is a great suggestion and thank you for pointing out the two field studies. We thought about testing the model in 1-D or 2-D modeling settings prior to implementing the LEVCHEM_v1 mechanism into a 3-D modeling framework, but this required an additional model development focus and setup that was beyond the goal of this study. However, we will still consider applying LEVCHEM_v1 to fire plumes in the future.

General Comments

*Reviewer comment:*

"In the paper it seems to go back and forth if it is written as gas-phase and aerosol-phase (with hyphens) or gas phase and aerosol phase (without hyphens). This should be checked throughout the entire text."

*Author comment:*

There are situations when the hyphen is needed and situations when the hyphen is not needed; this is not necessarily incorrect even if it appears to be inconsistent throughout the text. When we use it as an adjective (i.e. gas-phase reaction), we think that the hyphen is needed. When we use it differently (i.e., "it happens in the gas phase"), we do not think the hyphen is needed. Therefore, we made no changes in the text regarding this apparent issue.

Specific Comments

*Reviewer comment*

Line 73 – Suggest adding an of after developed use

*Author comment*

Thank you for pointing this out. In addition, the word "use" was not supposed to be there. Therefore, we corrected this part to read as "Here we developed a zero-dimensional (0-D) modeling framework…".

*Reviewer comment*

Line 173 – Suggest changing particle phase to particle-phase

*Author comment*

We did not follow this suggestion (see our motivation above).

*Reviewer comment*

Line 260 – The comma after Supplemental Information) should be a period

*Author comment*

We corrected this.

*Reviewer comment*

Line 340 – There is an extra space in gas-phase

*Author comment*

We corrected this.

*Reviewer comment*

Line 393 – In the Alvardo et al. reference, periods are missing in the initials for author Akagi

*Author comment*

We corrected this.

*Reviewer comment*

Line 394 – In the Arangio et al. reference, believe oh should be capitalized

*Author comment*

We corrected this.

*Reviewer comment*

Figure 3 -In the plot titles, the hyphen is missing in Gas-phase and Aerosol-phase
Figure 8 -In the plot titles, the hyphen is missing in Gas-phase and Aerosol-phase
Figures S3, S4, S5, S6 -In caption, the hyphen is missing in gas-phase and aerosolphase

*Author comment*

We accepted your suggestion for Figures 3 and 8 titles, because the use of the hyphen was justified in these cases (see section C). However, we do not think that a hyphen is needed in the captions of Figures S3 to S6 (and the new Fig. S7), for the reason we have already discussed (see our comment above).

**C. Other revisions**

*Note that in this section we only provide additional changes in the manuscript and supplemental information to support our answers to reviewers' comments (sections A and B). More changes will be explicitly addressed in the final Author Response, such as effects of LEV degradation on other gases (based on revisions of Figures 5-7) and any text updates regarding the new results and discussions.*

*Manuscript*

Partial revisions of the manuscript include:

Line revisions (lines correspond to those from unrevised manuscript)

Lines 207 and 214: We added "; Pratap et al., 2019"

Line 224: We replaced "5 hours" by "5-6 hours" because data from Pratap et al. (2019) extends beyond 5 hours.

Lines 224-226: We rephrased the sentence on these lines to include changes in F and α based on new simulations in response to reviewer's 1 major comment:

> "*Overall, the model predicted that LEV degradation closely follows the measured LEV degradation in relatively slower heterogeneous chemistry scenarios (F = 0.001; 0.002; 0.004; 0.02; 0.03, depending on the experimental data considered) and at mass accommodation coefficients of 0.1 and 0.01.*"

Line 274: Based on new simulations, we updated the range of SOA yields "5 to 32%" by "4 to 32%".

Line 315: We corrected the α value from 0.001 to 0.1 to reflect the conditions which were re-modeled in response to reviewer 1's major comment.

Line 313: We rephrased the sentence starting on this line to reflect the new results:

> "*Over these time scales, SOA yields vary roughly within the same range (14-33%) as observed in the previous cases considered (see section 3.2).*"

Figure revisions

Figure 3: We updated this figure (and its caption) based on our response to "Major comment" and "Suggestion" of reviewer 1 and suggestion from reviewer 2. It shows that degradation timescale of levoglucosan increases, particularly in the gas phase (from 1.5-3.5 days 1.5-5 days) but only because of the inclusion of Pratap et al. (2019) conditions in the simulations (purple dotted line). The effect of using much smaller $C_i^*$ values in the simulations is reflected by the larger α values (0.01-01 compared to 0.001) which were found to be effective, according to the data.

[Figure]

**Figure 3 Degradation of LEV (conditions from Hennigan et al. (2010) when F=0.02-0.03 and α = 0.1, from Lai et al. (2014) when F=0.002-0.004 and α = 0.01, and from Pratap et al. (2019) when F = 0.001 and α = 0.01 ). Note the change in the scale of the axes between the two panels.**

Figure 4: We updated this figure (and its caption) based on our response to the "Major comment" and "Suggestion" of reviewer 1. No significant changes compared to original Figure 4 are observed, except for the fact that Figure 4a additionally includes the modeled conditions from Pratap et al. (2019) (see the dotted purple line).

[Figure]

**Figure 4 (a) Evolution of SOA yields from LEV degradation using valid simulations (conditions from Hennigan et al. (2010) when F=0.02-0.03 and α = 0.1, from Lai et al. (2014) when F=0.002-0.004 and α = 0.01), and from Pratap et al. (2019) when F = 0.001 and α = 0.01). (b) Effect of varying the heterogeneous reaction rate coefficient by 4 orders of magnitude, at constant mass accommodation coefficient (α = 0.1) (conditions from Hennigan et al. (2010)).**

Figure 8: This figure (and its caption) was updated based on our response to the "Major comment" and "Suggestion" of reviewer 1 as well as based on revisions suggested by reviewer 2. Related to this figure, Figure 9 is presented here only to show that by using a smaller $C_i^*$ (1 µg m$^{-3}$), conditions simulated in Figure 8 also work at α = 1, thus satisfying the Kumala and Wagner (2001) criterion α ≥ 1 discussed in section A. The concentration in the aerosol phase changes slightly in magnitude, particularly in the slowest heterogeneous chemistry scenario (dashed red line), but degradation timescale is not affected in any phase.

[revised manuscript text omitted]

Table revisions

Table S1

We modified this table to include conditions from Pratap et al. (2019), as discussed in section A.

Table S1 Conditions used in chamber simulations for model evaluation

| Reference study | $LEV\_G_0$ (ppmv)[*] | $LEV\_A_0$ (ppmv) | $H_2O$ (ppmv) | RH (%) | $T$ (K) | $D_p$ (m) | $N_t$ (m$^{-3}$) | SAD (m$^{-1}$) | $\Delta H_{vap,i}$ (J mol$^{-1}$) | $\sigma$ (N/m) | $C_i^*(298\,K)$ (kg m$^3$) | $\rho$ (kg m$^3$) | D (m$^2$ s$^{-1}$) |
|---|---|---|---|---|---|---|---|---|---|---|---|---|---|
| Hennigan et al. (2010) | $9.62 \times 10^{-2}$ | $6.23 \times 10^{-3}$ | $2.22 \times 10^3$ | 10 | 293.15 | $5 \times 10^{-7}$ | $3.18 \times 10^7$ | $2.50 \times 10^{-5}$ | $84 \times 10^3$ | $2.82 \times 10^{-2}$ | $13 \times 10^{-9}$ | $1.69 \times 10^3$ | $5 \times 10^{-6}$ |
| Lai et al. (2014) | $9.62 \times 10^{-2}$ | $7.09 \times 10^{-4}$ | $1.18 \times 10^4$ | 40 | 298.15 | $2 \times 10^{-7}$ | $9.96 \times 10^8$ | $1.25 \times 10^{-4}$ | $84 \times 10^3$ | $2.82 \times 10^{-2}$ | $13 \times 10^{-9}$ | $1.69 \times 10^3$ | $5 \times 10^{-6}$ |
|  | $9.62 \times 10^{-2}$ | $7.09 \times 10^{-4}$ | $6.60 \times 10^3$ | 40 | 288.15 | $2 \times 10^{-7}$ | $9.96 \times 10^8$ | $1.25 \times 10^{-4}$ | $84 \times 10^3$ | $2.82 \times 10^{-2}$ | $13 \times 10^{-9}$ | $1.69 \times 10^3$ | $5 \times 10^{-6}$ |
| Pratap et al. (2019) | $9.62 \times 10^{-2}$ | $7.23 \times 10^{-3}$ | $3.63 \times 10^3$ | 30 | 283.15 | $2 \times 10^{-7}$ | $9.96 \times 10$ | $1.25 \times 10^{-4}$ | $84 \times 10^3$ | $2.82 \times 10^{-}$ | $13 \times 10^{-9}$ | $1.69 \times 10^3$ | $5 \times 10^{-6}$ |

---

## Author Response (AR1)

**Final author response to reviewers of the gmd-2020-189 manuscript**

We thank both reviewers for their constructive comments and helpful suggestions to improve our manuscript. Our responses to their comments are provided below, separately for each reviewer (sections A and B). At the end, we provide a third section (C), describing additional changes resulting from our answers to the reviewers' comments, followed by the revised manuscript with changes tracked.

**A. Reviewer 1 (Andrew May)**

Major comment

*Reviewer comment:*
"There appears to be a fundamental error regarding the volatility ($C_i*$) of levoglucosan in this modeling framework. Table S1 shows that $C_i* = 3 \times 10^{-3}$ kg m$^{-3}$ = $3 \times 10^6$ µg m$^{-3}$, which would imply that that vast majority of the levoglucosan should be in the gas phase (e.g., if COA = 30 µg m$^{-3}$, 99.999% would be in the gas phase). This does not match with the initial distribution of the gas-phase and aerosol-phase concentration in Table S1. Experimentally, we showed that this was more like 13 µg m$^{-3}$ in May et al. (2012). We speculated that this was why Hennigan et al. (2010) observed uptake coefficients (γ) greater than 1, and this could be a plausible explanation for the results in Knopf et al. (2011) for NO$_3$ radicals. Moreover, a structure-activity relationship (Mansouri et al., 2018) predicts a vapor pressure that yields $C_i* \approx 1$ µg m$^{-3}$. Perhaps this issue with $C_i*$ is the reason why α = 0.0001 is required to demonstrate good agreement in Figure 1?"

*Author comment:*
Indeed, the $C_i*$ value we used in our model is likely too large and we thank the reviewer for pointing this out. In our model we used the maximum value of a range from May et al. (2013). However, levoglucosan may have a $C_i*$ value closer to the lower end of that range, since it is not very volatile.

To address this, we ran new simulations by considering both suggestions ($C_i* = 13$ µg m$^{-3}$ and $C_i* = 1$ µg m$^{-3}$). We confirm that the model also predicts well the LEV degradation at larger α values than 0.001, respectively 0.01 and 0.1 (using $C_i* = 13$ µg m$^{-3}$) and 1.0 (using $C_i* = 1$ µg m$^{-3}$). The variations in the α response to $C_i*$ are driven by the different experimental conditions used to initialize the model. For example, using conditions from Hennigan et al. (2010), the model performed well at α = 0.1 ($C_i* = 13$ µg m$^{-3}$) and at α = 1.0 ($C_i* = 1$ µg m$^{-3}$). Conditions from Lai et al. (2014) worked well only at α = 0.01 ($C_i* = 13$ µg m$^{-3}$). However, we chose to report results only for cases when $C_i* = 13$ µg m$^{-3}$ was used because this value worked well with respect to α, model evaluation and model performance, for all the modeled scenarios, including the conditions from an additional experimental study that the reviewer has suggested (Pratap et al., 2019). These results are shown in revised Figures 1 and 2 below. Model performance was not negatively affected; on the contrary, the accuracy improved slightly (from 48% to 47%).

[Figure]

**Figure 1 Simulated LEV degradation (lines) and measured LEV degradation (points); Color represents conditions from different chamber experiments taken from three studies (red – Hennigan et al. (2010), blue – Lai et al. (2014) and green – Pratap et al. (2019)) used in the simulations. LEV concentration is normalized by the initial concentration (LEV/LEV$_0$).**

[Figure]

**Figure 2 Parity plot of predicted versus measured LEV concentration (normalized by the initial concentration). The dotted lines represent the ± 30% error margins.**

*Reviewer comment:*

"Related to the Major Comment, Kulmala and Wagner (2001) provide a theoretical relationship between mass accommodation coefficients (α) and γ. One key point that they make is that α ≥ γ (depending on the Knudsen number). In this work, γ >> α, based on Table 2. Therefore, the authors should either constrain α > 0.1 or provide some justification for α << γ. I suspect that this issue may work itself out once the Major Comment has been resolved, but it is worth noting in the event that it does not."

*Author comment:*

We confirm that by correcting the $C_i^*$ used in our model, we obtained good model predictions at α values that are 1-2 orders of magnitude larger compared to previous simulations at much larger $C_i^*$ values (see our response to the major comment above). These α values (0.01 and 0.1) are still smaller than those of the uptake coefficients (γ) for most of the chemical species, including for levoglucosan. However, as seen in Table 2, the great majority of the γ values were computed in this study, in the absence of their experimental measurements. In eq. 7, we assume a similar $2^{nd}$ order heterogeneous reaction rate for all the species (see line 165); this may bias our calculations of γ towards larger values. For γ values on the order of $10^{-1}$ (OH uptake by levoglucosan, for example) and Knudsen numbers on the order of $10^{-1}$ (all modeled cases), the corresponding α should be 0.1, according to Fig. 1 in Kulmala and Wagner (2001). This is true when we model conditions from Hennigan et al. (2010); thus, in this case the α ≥ γ criterion is satisfied. Modeled conditions from Lai et al. (2014) and Pratap et al. (2019) do not meet this criterion for levoglucosan because, for similar *Kn* and γ values, the model worked well (compared to experimental data) only at α = 0.01; in these cases, α < γ. However, for other species with smaller γ ($O_3$ and $N_2O_5$), all the modeled cases in our study satisfy the criterion α ≥ γ. It is worth noting here that the effective α values we found in our study, by comparing model predictions with data, have inherent uncertainties associated with both the data and the model.

To address this reviewer's comment directly in the manuscript:

(*i*) We introduce the relationship between α and γ described by Kulmala and Wagner (2001) by rephrasing the sentence on line 205:

> "*The mass accommodation coefficient is related to the G/P partitioning mechanism (eq. 14) and the uptake coefficient (γ). Theoretically, α ≥ γ, depending on the Knudsen number (Kulmala and Wagner, 2001).*"

*ii*) We discuss our results with respect to α and γ by adding the following paragraph starting on line 226:

> "*These α values are smaller than those of γ for most of the chemical species, including for levoglucosan. However, as seen in Table 2, the great majority of the γ values were computed in this study, in the absence of their experimental measurements. In eq. 7, we assume a similar $2^{nd}$ order heterogeneous reaction rate for all the species; this may bias our calculations of γ towards larger values. For γ values on the order of $10^{-1}$ (OH uptake by levoglucosan, for example) and Knudsen numbers on the order of $10^{-1}$ (all modeled cases), the corresponding α should be ~0.1, according to Fig. 1 in Kulmala and Wagner (2001). This is true when we model conditions from Hennigan et al. (2010); thus, in this case the α ≥ γ criterion is marginally satisfied. Modeled conditions from Lai et al. (2014) and Pratap et al. (2019) do not meet this criterion for levoglucosan because, for similar Kn and γ values, the model worked well (compared to experimental data) only at α = 0.01; in these cases, α < γ. However, for other species with smaller γ ($O_3$ and $N_2O_5$), all the modeled cases in our study satisfy the criterion α ≥ γ. It is worth noting here that the effective α values we found in our study by comparing model predictions with data have inherent uncertainties associated with both the data and the model.*"

Specific comments

*Reviewer comment:*

"Lines 198-203: As I'm reflecting on this after having read the full draft, it seems like these "F" values could be related to the mass fraction of levoglucosan in the particles. For example, F = 0.03 is the slope of "$m/z$ 60" (proxy for anhydrosugars in the Aerosol Mass Spectrometer world) vs. OA concentration in Figure 7 in Sullivan et al. (2014). I haven't carefully read through Lai et al. (2014), but conceivably, they could have used lower mass-mixing ratios of levoglucosan in their particles. I bring this up because as written, F essentially sounds like a "fudge factor" to get the model to agree with observations. Maybe something like "We expect F to be, at a maximum, 0.1 due to observed mass fractions in biomass burning organic aerosol""

*Author comment:*

This is an interesting suggestion. We used F precisely for accounting for the fact that there are many factors that may inhibit heterogeneous chemistry while the heterogeneous reaction rate ($k_{SFC\_REACTION}$) does not account for that. Model evaluation confirms that by not constraining $k_{SFC\_REACTION}$ by F, results are not realistic. We see that F could be a mass fraction accounting for levoglucosan (or other particle-phase species) that is available to react with gas-phase oxidants. At a minimum, we found F to be 0.001 (conditions from Pratap et al., 2019), while at a maximum, F was 0.02-0.03 (conditions from Hennigan et al., 2010) (see Figure 1 above). We also think that these variations in F are driven by different temperature (and possibly relative humidity) conditions modeled, with higher temperature sustaining larger F and lower temperature supporting lower F. Therefore, we added your suggestion starting on line 202 and we adjusted the next sentence to accommodate this addition as follows:

> *"We expect F to be, at a maximum, 0.1 due to observed mass fractions in biomass burning organic aerosols (Sullivan et al., 2014). However, for sensitivity analysis, we varied F from 1.0 (default case) to lower values (0.1, 0.01 and 0.001) to slow down the heterogeneous reaction rates."*

*Reviewer comment:*

"Lines 203-204: I have no recollection of making this claim regarding α in May et al. (2013), and quickly reviewing that paper, it appears that the closest thing to that is that there appeared to be no mass transfer limitations for the evaporation of biomass burning smoke. In May et al. (2012), we did include some discussion of this in the online supporting information, but that was more focused on the fact that even if we decrease α < 1, the evaporation timescale is shorter than heterogeneous chemistry timescale. I guess that my point here is that my previous work has been mis-interpreted.

These results, coupled with γ ≈ 1 (presumably, based on Kessler et al. (2010)?) and Kulmala and Wagner (2001) suggesting that α ≥ γ, seem to imply that α ≈ 1."

*Author comment:*

On Line 204, we refer to α as "…(which is 0.1 for a system in equilibrium (May et al., 2013)…". We inferred this from paragraph 18/page 11,330 (May et al., 2013), where α ≥ 0.1 is considered a good value for a system in equilibrium, as has been found in different chamber systems. We rephrased the sentence starting on line 203 as follows:

> *"In addition, we varied the mass accommodation coefficient (see eq. 14) from a default case of 0.1 (which is the lower limit of α for a system in equilibrium (May et al., 2013)) to lower values (0.01 and 0.001) and larger values (1.0). It was necessary to vary α because its value is unknown for levoglucosan and its degradation products."*

*Reviewer comment:*

"Section 3.2 and Figure 4: I am a bit confused by this discussion about yield, which is perhaps based on an *a priori* expectation that SOA yield increases with increased photochemical aging. Is the implication that initially, many of the oxidation products remain in the particle phase (t < ~ 6 hr) but as the chemistry continues, heterogeneous oxidation results in more volatile products until a steady-state is reached (at t ~ 24 -36 hrs?)"

*Author comment:*

If we look at Figures S3-S7 (see section C/Supplemental Information), about the same number of products (~3) is present in both phases, but the products differ between the two phases. Heterogeneous chemistry is the driver of these yields in the first hours. Condensation is negligible, since LEVP4, which is a product of gas-phase chemistry only, is not present in the aerosol phase; it only appears in the aerosol phase at the end of the simulation, in some cases (Fig. S4-S7). Evaporation is not important either since the products LEVP6 and LEVP7, generated by heterogeneous chemistry only, are not seen in the gas phase at all. Most of the oxidation products in the aerosol phase remain in the aerosol phase over the entire simulation period, except for LEVP5 and LEVP1 that may partition to the gas phase after 6 hours. The steady-state is reached because the concentration of precursors is near-zero and the oxidation products from heterogeneous chemistry and condensation (i.e., LEVP4 only) remain in the aerosol phase. So, yes, your interpretation is correct. Therefore, we supplemented this section by adding the following text starting on line 274 (for updated Figure 4, see section C/Manuscript):

> *"These high SOA yields in the first 6 hours are the result of rapid conversion of the precursors to aerosol-phase products, mainly due to heterogeneous chemistry. Because these products are not seen in the gas phase, evaporation does not influence the SOA yields in this early stage of the simulation; condensation of gas-phase products (LEVP4 and LEVP5) is also negligible (see Fig. S3-S7). Most of the oxidation products remain in the aerosol phase over the entire simulation period, except for LEVP5 and LEVP1 that may partition to the gas phase. SOA yield reaches steady-state at ~24-26 hours due to near-zero concentrations of the two precursors and the presence of oxidation products from heterogeneous chemistry and G/P partitioning (i.e., condensation of LEVP4) in the aerosol phase."*

Suggestion

*Reviewer comment:*

"Another dataset to consider would be that from Pratap et al. (2019). They also modeled their own data, explicitly accounting for gas-particle partitioning and chamber wall loss in addition to oxidation chemistry, in predicting particle-phase levoglucosan concentrations. If nothing else, these data can provide another valuable test set at colder temperatures than Hennigan et al. (2010) and Lai et al. (2014)."

*Author comment:*

We thank the reviewer for this suggestion. We considered it in our simulations (see revised Figs. 1 and 2 above). Because conditions of this study represent wintertime conditions (T = 10°C/283.15 K), the model line falls in the proximity of the model conditions from Lai et al. (2014) (T = 15°C/288.15 K) and works for an F of similar magnitude ($10^{-3}$) and α (0.01). We also refer to this study in the manuscript and Supplemental Information (see section C for additional changes). Simulation results based on this study are also included in Figs. 3 and 4 (see section C/Manuscript) and Fig. S7 (see section C/Supplemental Information).

**B. Reviewer 2 (anonymous)**

Suggestion

*Reviewer comment:*

"I would highly encourage the authors to try out their model on ambient levoglucosan data from aircraft campaigns that were able to follow smoke plumes over time such as WINTER and WECAN."

*Author comment*: This is a great suggestion and thank you for pointing out the two field studies. We thought about testing the model in 1-D or 2-D modeling settings prior to implementing the LEVCHEM_v1 mechanism into a 3-D modeling framework, but this required an additional model development focus and setup that was beyond the goal of this study. However, we will still consider applying LEVCHEM_v1 to fire plumes in the future.

General Comments

*Reviewer comment:*

"In the paper it seems to go back and forth if it is written as gas-phase and aerosol-phase (with hyphens) or gas phase and aerosol phase (without hyphens). This should be checked throughout the entire text."

*Author comment:*

There are situations when the hyphen is needed and situations when the hyphen is not needed; this is not necessarily incorrect even if it appears to be inconsistent throughout the text. When we use it as an adjective (i.e. gas-phase reaction), we think that the hyphen is needed. When we use it differently (i.e., "it happens in the gas phase"), we do not think the hyphen is needed. Therefore, we made no changes in the text regarding this apparent issue, except for the title of Table 1 (see the manuscript at the end of this author response).

Specific Comments

*Reviewer comment*

Line 73 – Suggest adding an of after developed use

*Author comment*

Thank you for pointing this out. In addition, the word "use" was not supposed to be there. Therefore, we corrected this part to read as "Here we developed a zero-dimensional (0-D) modeling framework…".

*Reviewer comment*

Line 173 – Suggest changing particle phase to particle-phase

*Author comment*

We did not follow this suggestion (see our motivation above).

*Reviewer comment*

Line 260 – The comma after Supplemental Information) should be a period

*Author comment*

We corrected this.

*Reviewer comment*

Line 340 – There is an extra space in gas-phase

*Author comment*

We corrected this.

*Reviewer comment*

Line 393 – In the Alvardo et al. reference, periods are missing in the initials for author Akagi

*Author comment*

We corrected this.

*Reviewer comment*

Line 394 – In the Arangio et al. reference, believe oh should be capitalized

*Author comment*

We corrected this.

*Reviewer comment*

Figure 3 -In the plot titles, the hyphen is missing in Gas-phase and Aerosol-phase

Figure 8 -In the plot titles, the hyphen is missing in Gas-phase and Aerosol-phase

Figures S3, S4, S5, S6 -In caption, the hyphen is missing in gas-phase and aerosolphase

*Author comment*

We accepted your suggestion for Figures 3 and 8 titles, because the use of the hyphen was justified in these cases (see section C). However, we do not think that a hyphen is needed in the captions of Figures S3 to S6 (and the new Fig. S7), for the reason we have already discussed (see our comment above).

**C. Other revisions**

In this section, we explicitly provide additional changes in the Manuscript and Supplemental Information resulting from our answers to reviewers' comments (see sections A and B).

**Manuscript**

Revisions of the manuscript include:

Line revisions (lines correspond to those in the original manuscript)

Line 10: We removed the extra space between "of" and "a".

Line 16: We added "gas-phase" between "with" and "degradation".

Line 17: We replaced "3.5 days" by "5 days" and "8-21 hours" by "8-36 hours".

Lines 18-20: We rephrased this sentence to point out the conditions tested in the sensitivity analysis and to adjust the text in response to the change made on line 17:

> "*We varied the heterogeneous reaction rate constant in a sensitivity analysis (for summer conditions only) and found that longer degradation time scales of LEV are possible, particularly in the aerosol phase (7 days), implying that some LEV may be transported regionally.*"

Line 25: Based on new simulations, we updated the range of SOA yields "5 to 32%" by "4 to 32%".

Line 28: We deleted "(VOC)".

Line 30: Because we modeled experimental conditions from an additional study (Pratap et al., 2019), which extend to 5.6 hours, we replaced "3-5 hours" by "3-6 hours".

Line 56: We added "; Pratap et al., 2019" to the references cited on this line.

Line 57: We replaced "character" by "chemistry".

Line 60: To acknowledge the previous works of Pratap et al. (2018, 2019) with respect to modeled gas-particle partitioning and multiphase chemistry of levoglucosan, we added the following sentence starting on this line:

> "Previous studied applied the gas-particle partitioning model of May et al. (2013) to levoglucosan but its multiphase chemical decay was limited to the reaction with the OH radical only (Pratap et al., 2018; 2019)."

Lines 61 and 325-326: We corrected "modelled" by "modeled".

Line 68: We added a hyphene to the word gas.

Line 82: We corrected "form" by "from".

Line 112: We replaced "outputs" by "information" to avoid using the same word twice in the sentence.

Line 148: We deleted the "or surface area".

Line 191: We deleted "organic molecules in".

Lines 207 and 214: We added "; Pratap et al., 2019"

Line 213: We deleted "one".

Line 224: We replaced "5 hours" by "5-6 hours" because data from Pratap et al. (2019) extends to 5.6 hours.

Lines 224-226: We rephrased the sentence on these lines to include changes in F and α based on new simulations in response to reviewer 1's major comment:

> "*Overall, the model predicted that LEV degradation closely follows the measured LEV degradation in relatively slower heterogeneous chemistry scenarios (F = 0.001; 0.002; 0.004; 0.02; 0.03, depending on the experimental data considered) and at mass accommodation coefficients of 0.1 and 0.01.*"

Line 229: We added "Pratap et al. (2019)".

Line 235: We corrected "versus" from italic to non-italic.

Lines 236-237: We corrected the sentence to include the modeled scenario in green "scenarios (red, blue and green)".

Lines 238-239: Based on new simulations, we corrected "48%" by "47%".

Line 241: We replaced "five" by "5-6", since data from Pratap et al. (2019) extends slightly beyond 5 hours.

Lines 242-243: Based on new simulations, we replaced "1.5-3.5 days" by "1.5-5 days" and "8-21 hours" by "8-36 hours". We also added on line 242 the word "nearly" since the concentration of LEV was not zero after this time scales.

Lines 246-247: We replaced "5 hours" by "5-6 hours".

Line 247: We replaced "Figure S4" by "Figure S7".

Line 254: We replaced "5 hours" by "5-6 hours".

Lines 259-264: We deleted the text on these lines as we found it not so relevant.

Line 365: We replaced "nitrogen cycle" by "nitrogen cycling" and "carbon cycle" by "carbon cycling".

Line 273: We replaced "1-12 minutes" by "2-34 minutes".

Line 274: Based on new simulations, we updated the range of SOA yields "5 to 32%" by "4 to 32%".

Lines 274-277: Based on new simulations, we revised the text on these lines as follows:

"*Among the simulated scenarios, the largest SOA yields resulted when higher initial LEV_A concentrations were used in the simulations and they did not decrease below 8% in wintertime conditions (Figure 4a and Table S1). The heterogeneous chemistry was the slowest for SOA yields predicted for winter conditions (suggested by F = 0.001) while it was the fastest for those associated with summer conditions (F = 0.02-0.03).*"

Lines 277-278: Based on new simulations, we corrected the total aerosol mass described on these lines as follows:

"*The total aerosol mass (the sum of concentrations of all LEV-related aerosol species, including the radicals) also increased by 8-15% in the first six hours and kept increasing, although at a slower pace, to up to 18-29% at the end of the simulation period. The smallest total aerosol mass in the first six hours (8%) was observed in modeled wintertime conditions, while the highest total aerosol mass (14-15%) was observed in summertime conditions.*"

Line 278: We adjusted the beginning of the sentence starting on this line as "These suggest that…".

Lines 302-303: Based on new simulations, we corrected the linear slopes of $O_3$ vs. $NO_x$ and $O_3$ vs. VOC as follows:

"…(linear slope of $-0.250 \pm 0.001$ ppb/ppbC) but not as much as $NO_x$ did (linear slope of $-2.821 \pm 0.007$ ppb/ppb)…".

Line 304: Based on new simulations, we corrected the concentration of $O_3$ mentioned on this line from "117 ppb" to "112 ppb".

Line 315: We corrected the α value from 0.001 to 0.1 to reflect the conditions which were re-modeled in response to reviewer 1's major comment.

Lines 316-318: Based on new simulations, the comparison made on these lines does hold only for aerosol-phase levoglucosan. Therefore, we reformulated the sentence as:

"*While the time scale of gas-phase LEV is similar (5 days) to that observed with reaction rates used in chamber comparisons (see section 3.1), the time scale of aerosol-phase LEV is much larger (7 days versus 36 hours), suggesting that LEV associated with PM can be transported and deposited regionally.*"

Line 313: We rephrased the sentence starting on this line to reflect the new results:

"*Over these time scales, SOA yields vary roughly within the same range (14-33%) as observed in the previous cases considered (see section 3.2).*"

Lines 322-326: Based on new simulations, we also adjusted the text on these lines as follows:

"*We also tested the sensitivity of the mass accommodation coefficient (α) at F = 0.01, using conditions from Hennigan et al. (2010). Varying α by four orders of magnitude (0.001, 0.01, 0.1 and 1.0) showed little effect on LEV degradation (i.e., degradation in the gas phase was slightly faster when α = 1, while degradation in the aerosol phase was slightly faster when α = 0.001-0.1) in comparison to the effect of slowing down the heterogeneous chemistry (F, as described above). The effect of the mass accommodation coefficient on LEV degradation appears to be more important when the G/P partitioning is modeled as gas-aerosol equilibrium reactions of which the partitioning coefficient is modeled with eq. 13.*"

Lines 337-343: Based on new simulations, we adjusted the text on these lines as follows:

"*We found that LEV degradation time scale ranges from 8-36 hours (aerosol-phase) to 1.5-5 days (gas-phase); however, model output was evaluated only for six hours through comparison to chamber measurements. In addition, we conducted a sensitivity analysis investigating a factor slowing down the heterogeneous chemistry and found that longer degradation time scales may occur, particularly in the aerosol phase (7 days). This longer time scale is slightly larger*

*than that of deposition (1-5 days) but is slightly shorter than that of regional transport (10 days), suggesting that some fraction of aerosol-phase LEV may be transported regionally.*"

Lines 347-352: Based on new simulations, we updated the text on these lines to reflect the new findings as follows:

"*LEV degradation contributes to SOA formation that was quantified mainly through simulated SOA yields. Based on 6-h degradation time scales, simulated SOA yields ranged from 4 to 32% and peaked in the first 2-34 minutes. Varying the heterogeneous chemistry rate by four orders of magnitude did not result in significantly different SOA yields (14-33%). The total PM mass (determined as the ratio of total aerosol concentration to initial LEV_A concentration) increased by 8-15% in the first six hours of all simulations and continued to slowly increase to 18-29% at the end of the simulation period.*"

Line 359: We replaced "117 ppb" by "112 ppb".

Line 383: We replaced the link to the source of LEVCHEM_v1 model files "https://zenodo.org/record/3885786" by "https://zenodo.org/record/4215973". This new link stores the model files representing simulation setups, inputs and outputs of the new simulation runs in response to reviewer's 1 major comment.

Reference revisions

We added four new references to this section:

Kulmala, M. and Wagner, P. E.: Mass accommodation and uptake coefficients - a quantitative comparison, J. Aerosol Sci., 32, 7, 833–841, https://doi.org/10.1016/S0021-8502(00)00116-6, 2001.

May, A. A., Saleh, R., Hennigan, C. J., Donahue, N. M. and Robinson, A. L.: Volatility of organic molecular markers used for source apportionment analysis: measurements and implications for atmospheric lifetime, Environ. Sci. Technol., 46, 22, 12435-12444, https://doi.org/10.1021/es302276t, 2012.

Pratap, V., Chen Y., Yao G. and Nakao S.: Temperature effects on multiphase reactions of organic molecular markers: A modeling study, Atmos. Environ., 179, 40-48, https://doi.org/10.1016/j.atmosenv.2018.02.009, 2018.

Pratap, V., Bian, Q., Kiran, S. A., Hopke, P. K., Pierce, J. R. and Nakao, S.: Investigation of levoglucosan decay in wood smoke smog-chamber experiments: The importance of aerosol loading, temperature, and vapor wall losses in interpreting results, Atmos. Environ., 199, 224–232, https://doi.org/10.1016/j.atmosenv.2018.11.020, 2019.

We corrected most of the references by: (1) adding "and" before the last author, (2) adding space between author initials and (3) adding the period to the abbreviated journal names.

Figure revisions

Figure 3: We updated this figure (and its caption) based on our response to "Major comment" and "Suggestion" of reviewer 1 and suggestion from reviewer 2. It shows that degradation timescale of levoglucosan increases in both phases (gas-phase: from 1.5-3.5 days to 1.5-5 days; aerosol-phase: from 8-21 hours to 8-36 hours) but only because of the inclusion of Pratap et al. (2019) "wintertime" conditions in the simulations (purple dotted line). The effect of using much smaller $C_i^*$ values in the simulations is reflected by the larger α values (0.01-01 compared to 0.001) which were found to be effective, according to the data.

[Figure]

**Figure 3 Degradation of LEV (conditions from Hennigan et al. (2010) when F=0.02-0.03 and α = 0.1, from Lai et al. (2014) when F=0.002-0.004 and α = 0.01, and from Pratap et al. (2019) when F = 0.001 and α = 0.01 ). Note the change in the scale of the axes between the two panels.**

Figure 4: We updated this figure (and its caption) based on our response to the "Major comment" and "Suggestion" of reviewer 1. No significant changes compared to original Figure 4 are observed, except for the fact that Figure 4a additionally includes the modeled conditions from Pratap et al. (2019) (see the dotted purple line).

[Figure]

**Figure 4 (a) Evolution of SOA yields from LEV degradation using valid simulations (conditions from Hennigan et al. (2010) when F=0.02-0.03 and α = 0.1, from Lai et al. (2014) when F=0.002-0.004 and α = 0.01), and from Pratap et al. (2019) when F = 0.001 and α = 0.01). (b) Effect of varying the heterogeneous reaction rate coefficient by 4 orders of magnitude, at constant mass accommodation coefficient (α = 0.1) (conditions from Hennigan et al. (2010)).**

Figures 5-7: The plotted information in these figures did not change significantly. However, because we re-ran simulations in response to reviewer's 1 comment (see section A) and included conditions from an additional data set (Pratap et al., 2019), we re-plotted the results in these figures (see below) and replaced the figures in the original manuscript.

[revised manuscript text omitted]

Table revisions

Table S1

We modified this table to include conditions from Pratap et al. (2019), as discussed in section A.

Table S1 Conditions used in chamber simulations for model evaluation

[revised manuscript text omitted]